# Alternative splicing modulation by G-quadruplexes

Ilias Georgakopoulos-Soares[1,2,11], Guillermo E. Parada [1,3,4,8,9,11], Hei Yuen Wong[5], Ragini Medhi[3,4], Giulia Furlan[3,4], Roberto Munita [6], Eric A. Miska[1,3,4], Chun Kit Kwok[5,7] & Martin Hemberg [1,3,10 ✉]

Alternative splicing is central to metazoan gene regulation, but the regulatory mechanisms are incompletely understood. Here, we show that G-quadruplex (G4) motifs are enriched ~3-fold near splice junctions. The importance of G4s in RNA is emphasised by a higher enrichment for the non-template strand. RNA-seq data from mouse and human neurons reveals an enrichment of G4s at exons that were skipped following depolarisation induced by potassium chloride. We validate the formation of stable RNA G4s for three candidate splice sites by circular dichroism spectroscopy, UV-melting and fluorescence measurements. Moreover, we find that sQTLs are enriched at G4s, and a minigene experiment provides further support for their role in promoting exon inclusion. Analysis of >1,800 high-throughput experiments reveals multiple RNA binding proteins associated with G4s. Finally, exploration of G4 motifs across eleven species shows strong enrichment at splice sites in mammals and birds, suggesting an evolutionary conserved splice regulatory mechanism.

[1] Wellcome Sanger Institute, Wellcome Genome Campus, Hinxton CB10 1SA, UK. [2] Department of Bioengineering and Therapeutic Sciences, University of California San Francisco, San Francisco, CA 94158, USA. [3] Wellcome Cancer Research UK Gurdon Institute, University of Cambridge, Tennis Court Road, Cambridge CB2 1QN, UK. [4] Department of Genetics, University of Cambridge, Downing Street, Cambridge CB2 3EH, UK. [5] Department of Chemistry and State Key Laboratory of Marine Pollution, City University of Hong Kong, Kowloon Tong, Hong Kong SAR, China. [6] Division of Molecular Hematology, Department of Laboratory Medicine, Lund Stem Cell Center, Faculty of Medicine, Lund University, Lund, Sweden. [7] Shenzhen Research Institute of City University of Hong Kong, Shenzhen, China. [8] Present address: Donnelly Centre for Cellular and Biomolecular Research, University of Toronto, Toronto, ON M5S 3E1, Canada. [9] Present address: Department of Molecular Genetics, University of Toronto, Toronto, ON M5A 1A8, Canada. [10] Present address: Evergrande Center for Immunologic Diseases, Harvard Medical School and Brigham and Women's Hospital, Boston, MA 02115, USA. [11] These authors contributed equally: Ilias Georgakopoulos-Soares, Guillermo E. Parada. ✉email: mhemberg@bwh.harvard.edu

In eukaryotes, pre-mRNA processing is key to gene regulation and the generation of isoform diversity. Alternative splicing is arguably the most pivotal mRNA processing mechanism in higher eukaryotes, and in humans it contributes substantially to protein diversity, affecting 95% of mRNA transcripts[1–4]. Moreover, alternative splicing is essential for normal cell growth, cell death, differentiation, development, sex, circadian rhythms, responses to environmental changes and pathogen responses[5–7].

The accuracy of pre-mRNA splicing relies on the recognition of three core signals; the 5′ splice site (5′ss), the 3′ splice site (3′ss), and the branch point. Despite the high fidelity observed during the splicing process, computational analyses have reported that human splice site core signals contain only half of the information required to accurately define exon/intron boundaries, implying the involvement of additional sequence features in splice site selection[8,9]. Some of the additional information necessary for splice site definition is found in a complex combination of *cis*-regulatory elements. These splice regulatory elements are short nucleotide sequences that are often bound by RNA-binding proteins (RBPs) that can either facilitate or inhibit the splice site recognition. The role of RBPs and splicing enhancers has been extensively studied, and the current understanding goes a long way towards a quantitative, predictive model of alternative splicing[10,11].

In addition to RBPs, secondary RNA structures are known to modulate alternative splicing[12], yet little is known about the impact of DNA secondary structures over alternative splicing. More than 20 non-canonical secondary structures have been previously reported for DNA[13], including G-quadruplexes (G4s), hairpins, cruciforms and triplexes. Sequences that predispose the DNA to non-canonical conformations are known as non-B DNA motifs, and they have been characterised with respect to their roles in gene regulation. It has been demonstrated that non-B DNA motifs can influence several aspects, including transcription initiation, transcription termination, and translation initiation[14–20]. Among the non-canonical secondary structures, G4s are the most widely studied class as they have been reported to have an important role in the transcriptional regulation of clinically relevant genes. For example, a DNA G4 in the promoter of the oncogene *MYC* acts as a repressor[21–23]. Similarly, a DNA G4 in the promoter of the proto-oncogene *KRAS* has a negative effect on expression levels[24]. Moreover, DNA G4s are also implicated in genomic instability in cancer and neurodegenerative diseases[21,25–28].

Many non-B DNA motifs will result in similar secondary structures at the RNA level[29–31]. In particular, abundant RNA G4 structure formation in the transcriptome has been demonstrated recently[32]. Importantly, the impact of RNA secondary structures in alternative splicing remains only partially understood[33,34] and although a role of G4s in splicing has been suggested[35–41] the extent of G4 impact on alternative splicing remains to be explored. Here we provide a genome-wide characterisation across multiple species of the role of non-B DNA motifs in alternative splicing.

## Results

### Sequence analysis and experimental data show that DNA G4s are enriched near splice sites.

To investigate the contribution of non-canonical secondary structures to splice site definition, we systematically explored the distribution of seven known non-B DNA motifs. Since the secondary structures can form both at the DNA and the RNA level[29,31,42], we initially considered both DNA strands. These motifs can be identified from the primary sequence, and we focused on the regions flanking human splice sites (Methods). The enrichment profiles varied substantially across the different non-B DNA motif categories (Fig. 1A), with exon-intron junctions displaying an acute enrichment for G4s,

short tandem repeats and H-DNA motifs. The high enrichment of short tandem repeats was expected since a subset of them overlap with intronic polypyrimidine tracts, which are known to be part of the core splicing signal[43,44]. By contrast, the enrichment patterns for G4s or H-DNA motifs cannot be explained by the distribution of known splicing signals.

The highest enrichment was for G4 motifs, both at the 3′ss (2.44-fold) and the 5′ss (4.06-fold), and this prompted us to further investigate if they have a role in the regulation of splicing. It has been shown previously that GC content is higher in exonic regions[45,46], but to control for the effect of the nucleotide composition of splice sites in the distribution of the GC-rich G4 motifs, we shuffled the 100 nt window on each side of the splice site while controlling for dinucleotide content. Comparing the observed frequency to the median from 1000 permutations we observed a corrected 2.53-fold and 2.73-fold enrichment for the frequency of G4 motifs at the 3′ss and 5′ss, respectively ($p$ value < 0.001 in both 3′ss and 5′ss), indicating that the G4 patterns are not driven by the sequence composition of splice sites. Moreover, the enrichment was consistent between human and mouse splice sites (Fig. 1A and Supplementary Fig. 1a), and the colocalization of G4 motifs and splice sites are not driven by a small number of loci. Within 100 nt of each splice junction, we identified 19,987 and 20,088 G4 motifs at the 3′ss and 5′ss, respectively. In total, 31% of human genes contain a G4 motif near at least one splice site within a distance of 100 bp. G4 motifs were found within 100 nt for 8.79% and 8.83% of the 3′ss and 5′ss, respectively. The reported G4 motif frequencies are likely a conservative estimate since we do not take into account intermolecular G4s or G4s that do not adhere to the consensus motif (G ≥ 3N1-7G ≥ 3N1-7G ≥ 3N1-7G ≥ 3)[47–49].

To evaluate if the G4 motifs that were enriched near splice sites lead to the formation of DNA G4 structures in vitro, we analysed previously published G4-seq data[50,51]. G4-seq utilises the fact that stable G4s can stall the DNA polymerase in vitro, thereby allowing high-throughput sequencing to be used to detect DNA G4s at high resolution. Chambers et al. provided the first method that enabled genome-wide detection of sites with DNA G4 formation potential, and they identified non-canonical structural features of DNA G4 formation as well as regions in the genome that are more likely to harbour DNA G4s, such as 5′ untranslated regions and splicing sites. We first measured the distribution of DNA G4s relative to the splice sites for HEK-293T cells in Pyridostatin (PDS) and K⁺ treatments from Marsico et al.[51]. PDS is a highly potent small molecule that binds and stabilises G4s. PDS, K⁺ and Na⁺ molecules selectively interact with G4s and stabilise them[52]. Compared to Na⁺, K⁺ stabilises G4 assemblies to a larger extent. In both conditions, we observed an enrichment of G4-seq peaks relative to the 3′ss and 5′ss, but with a more pronounced DNA G4 enrichment in PDS treatment compared to K⁺ treatment (Fig. 1B). The majority of DNA G4 positions derived from G4-seq peaks in K⁺ and PDS treatments did not overlap consensus G4 motifs (Fig. 1C). Since the G4-seq assay can also identify DNA G4s with non-canonical motifs, it is to be expected that the overlap with the consensus G4 motifs would be limited. We replicated our results in primary human B lymphocytes (NA18507) in Na⁺-K⁺ and Na⁺-PDS conditions[50], both of which promote G4 formation. In both conditions, only ~25% of G4-seq derived peaks are captured by the consensus G4 motif. Nevertheless, at splice sites we found an enrichment comparable to that obtained from the motif analysis, directly implicating G4 formation at splice sites (Supplementary Fig. 1b–d). Differences between the G4-seq datasets are likely the result of the differences in the experimental settings and treatments between the two studies[50,51]. For both the PDS and K⁺ treatments, we find that a substantial fraction of the genome

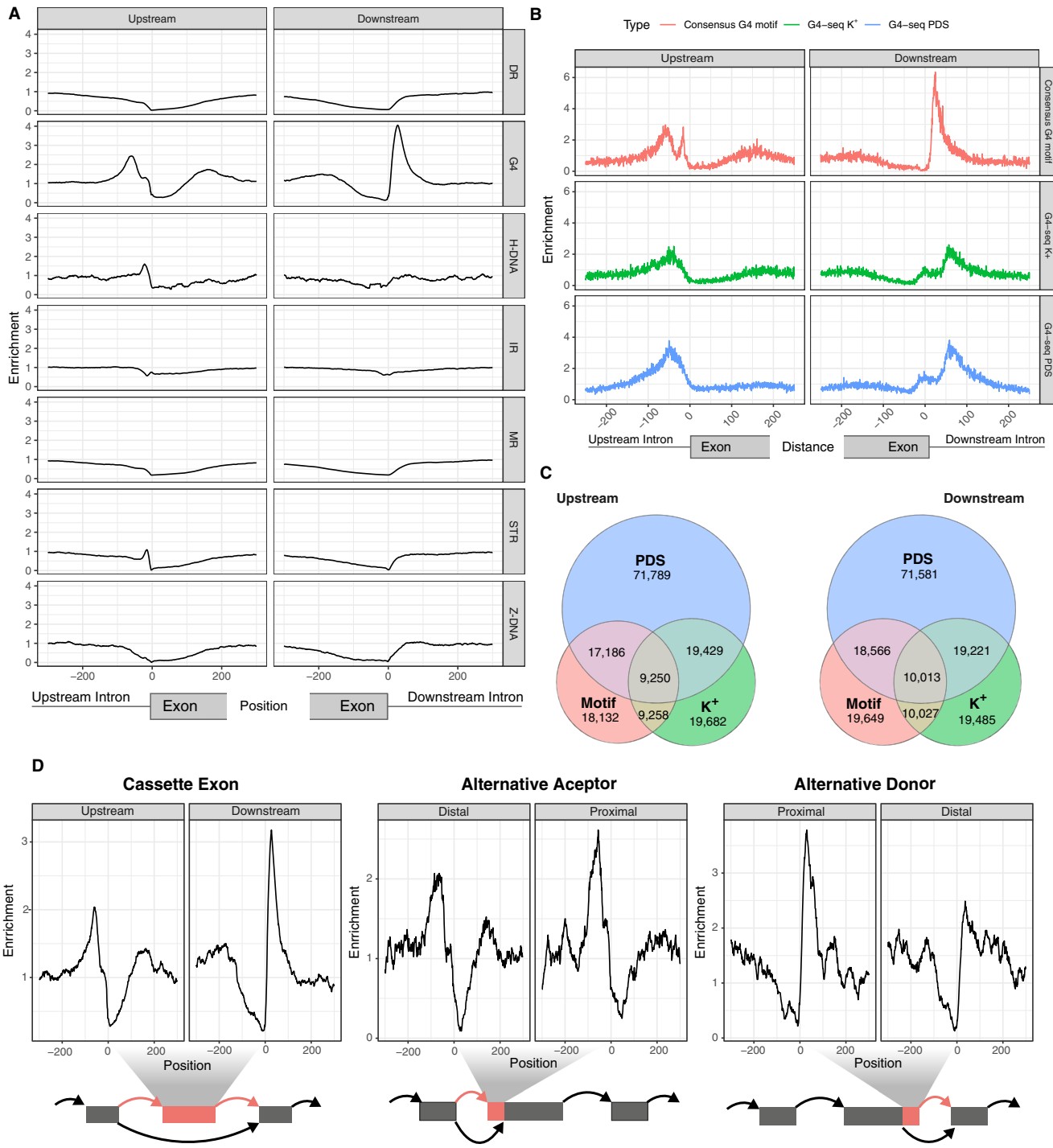

**Fig. 1 Non-B DNA motifs at splicing junctions. A** Distribution of non-B DNA motifs relative to splice sites. Seven non-B DNA motifs are shown, namely direct repeats (DRs), G-quadruplexes (G4s), H-DNA, inverted repeats (IRs), mirror repeats (MRs), short tandem repeats (STRs) and Z-DNA. **B** Distance between nearest G4 motif/G4-seq peak and a splice site separately for 3′/5′ splice sites. **C** Venn diagrams for the occurrences of G4s within 100 nt of the 3′ss (upstream) and 5′ss (downstream) using the consensus G4 motif, the K+ treatment G4-seq derived DNA G4 peaks and the PDS treatment G4-seq derived DNA G4 peaks[51] and reporting the overlapping G4s between them. **D** Distribution of G4 motifs relative to cassette exons, alternative acceptor and alternative donor. The orange colour in the schematic represents the alternatively included exonic part, corresponding to each type of alternative splicing event.

is affected, with 31.72 and 10.25% of splice junctions having a G4 within 100 bp. In addition, 67 and 35% of human genes contain a G4-seq peak from PDS and K+ treatments within 100 bp of a splice junction, supporting our earlier observations using the consensus G4 motif. As a result of these findings, we conclude that DNA G4s are a pervasive feature across splicing junctions.

**DNA G4 distribution patterns are found across splice site categories.** We extracted five different types of splicing sites (exon skipping, intron retention, alternative donor, alternative acceptor and mutually coordinated events) from VastDB[53], a curated alternative splicing database. We analysed the enrichment profile of G4 motifs across the different types of alternative splicing

events, and we found distinct differences (Fig. 1D and Supplementary Fig. 1e–g). Cassette exons are the most common type of alternative splicing event[54], and we found an enrichment profile for G4s consistent with our previous results (Fig. 1A, D). Interestingly, for both alternative acceptor and alternative donor events, we found that the enrichment was higher for the proximal than for the distal sites (Fig. 1D). These results provide evidence that G4s are associated with multiple splice site categories.

**DNA G4s are preferentially found on the non-template strand**. Thus far, we have analysed G4 sequences at the DNA level, but following transcription some of these could lead to RNA G4 formation. Since G4 motifs are strand-specific, we oriented each instance relative to the direction of transcription. Thus, we considered DNA G4s found at the template (non-coding) and non-template (coding) strands separately and found them statistically enriched on the non-template strand (Binomial tests, $p$ value < 0.001 at 3′ss and 5′ss). DNA G4s were enriched at both strands. At 3′ss the enrichment was 3.01-fold and 2.78-fold enrichment scores at the non-template and template strands, respectively (Supplementary Fig. 2). At the 5′ss the difference between the strands was larger with 5.56-fold and 2.38-fold at the non-template and template strands, respectively (Supplementary Fig. 2). Therefore, there was an asymmetric enrichment between the template and non-template strands at the 5′ss, but only a weak asymmetry at the 3′ss.

**DNA G4s are enriched at weak splice sites**. Weak splice sites are highly involved in alternative splicing and often contain additional regulatory elements[55–57]. To explore the distribution of G4s across weak and strong splice sites, we calculated a splicing strength score for all internal exons based on splice site position weight matrices[55,58]. We grouped splice sites into four quantiles based on the splicing strength scores and explored the enrichment levels of G4 motifs for each quantile separately. We found an inverse relationship between the calculated splicing strength score and G4 enrichment, with the weakest splice sites having the highest enrichment of G4s both at the 3′ss and the 5′ss with 2.77-fold and 4.95-fold enrichment, respectively (Supplementary Fig. 3). For both mouse and human, the splicing strength scores for splice junctions with a G4 are significantly lower than for splice junctions without a G4 (Mann–Whitney $U$, $p$ value <0.001). The same inverse relationship between the splicing strength score at splice sites and the DNA G4 enrichment is found in the G4-seq data for human in PDS and K[+] treatment[51] and for PDS-Na[+] and K[+]-Na[+] treatment[50] (Supplementary Fig. 4a–d), (Mann–Whitney $U$, $p$ values < 0.001).

We also investigated if there was a strand asymmetry when considering the splicing strength scores. Indeed, we found a bias in the splicing strength scores dependent on the strand orientation of G4s (Mann–Whitney $U$, 3′ss $p$ value <0.05, 5′ss $p$ value < 0.001). At the 3′ss the enrichment for splice junctions with the weakest splicing strength scores at the template and non-template strand was 3.90-fold and 3.66-fold, respectively. By contrast, we observed a 6.76-fold enrichment for G4s at the 5′ss at the non-template strand, but only a 3.66-fold enrichment on the template strand at the splice junctions with the weakest splicing strength scores (Fig. 2A). Taken together, these results indicate a preference for the non-template strand that is inversely proportional to the splicing strength score (Fig. 2A). They also suggest that G4s are more prevalent at the non-template strand, which is the sequence found in the transcribed mRNA. We validated the differences in the distribution of DNA G4 motifs at the template and non-template strands and the associated differences in

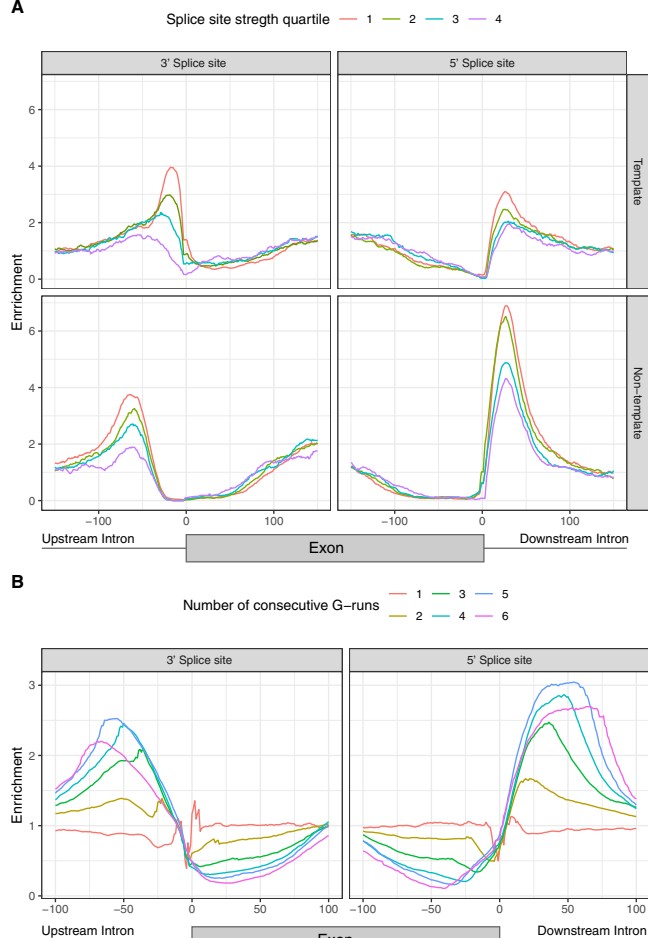

**Fig. 2 Characterisation of DNA G4 positioning across splicing junctions.** **A** G4 enrichment for template and non-template strands and stratified by the splicing strength scores of the adjacent splice site. The splicing strength scores for splice junctions with a G4 are significantly lower than for splice junctions without a G4 (Mann–Whitney $U$, $p$ value <0.001). The splicing strength score bias was found to be dependent on the strand orientation of G4s for splice sites with G4s within 100 nt away (Mann–Whitney $U$, 3′ss $p$ value <0.05, 5′ss $p$ value <0.001). **B** Number of consecutive G-runs and relative enrichment at the splicing junction. The error bands in **A**, **B** represent 95% confidence intervals from the binomial error.

splicing strength score using G4-seq from the two available datasets[50,51]. Consistent with the results derived from the consensus G4 motif, we found that DNA G4 formation has a preference for the non-template strand at the 5′ss of weak splice sites using the G4-seq datasets (Supplementary Fig. 4e–h).

**Longer G-runs are more highly enriched at splice sites**. An intramolecular G4 is usually a representation of four or more consecutive G-runs. Yet, intermolecular G4s can form with fewer G-runs since multiple molecules can contribute to G4 structure formation[59]. We found minimal to no enrichment for single G-runs at both 5′ss and 3′ss (Fig. 2B). However, for two and three G-runs we observed a 1.39-fold and a 2.10-fold enrichment at the 3′ss and a 1.67-fold and a 2.47-fold enrichment at the 5′ss, which may implicate intermolecular G4s in splice sites. The highest enrichment was observed for four to six G-runs, indicating that intramolecular G4 motifs are more enriched at splice sites than their intermolecular counterparts.

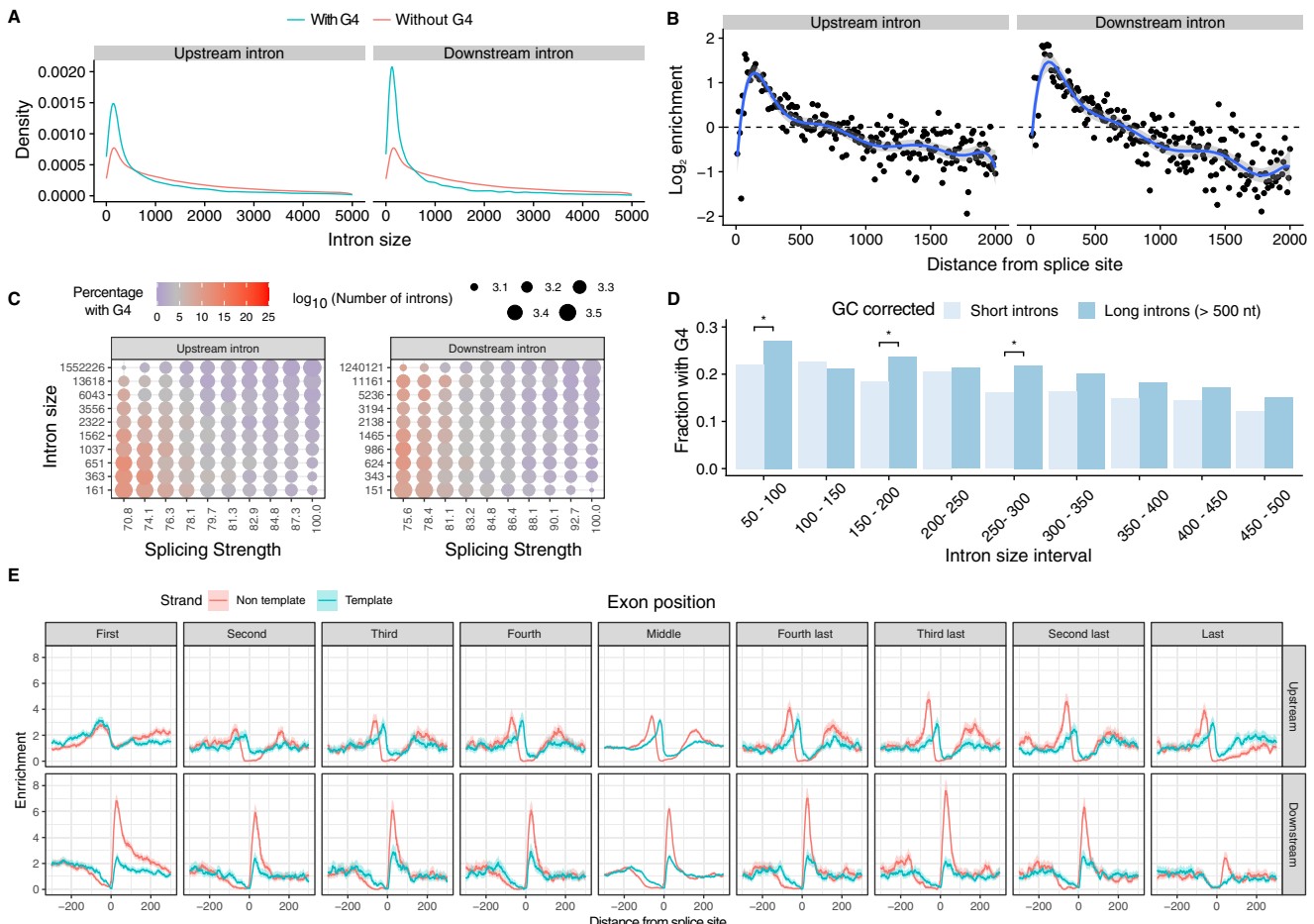

**Fig. 3 Distribution of DNA G4s in the vicinity of splicing sites. A** Length density distribution of introns upstream and downstream from exons that are flanked or not by G4s (two-sided Kolmogorov–Smirnov test *p* value <0.001). **B** Abundance enrichment of intron sizes at upstream and downstream splice sites flanked by G4s. A bin size of 10 bps was used with the blue line representing an eighth-degree polynomial model. Error bands represent 95% confidence intervals of the regression model. **C** Heatmap for the relationship between splicing strength score, intron length and G4 presence in a local window of 100 nt within the splice site for the upstream and downstream introns. Red colour represents a high proportion of splice site regions with G4s, whereas blue colour represents depletion of G4s. **D** Fraction of splice sites with a G4, controlling for GC content between long and short introns. We use chi-squared test to evaluate significant differences between short and long introns (* denotes *p* values <0.05 after multiple testing corrections). **E** G4 motif enrichment relative to the splice site across exons in the gene body at the 3′ss and at the 5′ss for template and non-template strands. G4 motifs are enriched at both 3′ss and 5′ss across splice sites throughout the gene body. Exons were separated into first to fourth exons, middle exons, last four exons and the distribution of G4s were studied individually for each category. Error bands represent 95% confidence intervals based on the binomial distribution.

**DNA G4s are enriched for short introns**. The length of introns in metazoans can vary across four orders of magnitude[60]. We hypothesised that the enrichment patterns of G4s at introns proximal to splicing sites would be associated with intron length. We compared the intron length of splice sites that had a G4 motif within 100 bps in the direction of the intron to the ones that did not have this motif. Consistent with our hypothesis, we found that introns with a G4 at the 3′ss had a median length of 701 nt while introns without a G4 had a median length of 1618 nt (Supplementary Fig. 5a), (Mann–Whitney *U*, *p* value < 0.001). Similarly, at the 5′ss, introns with a G4 had a median length of 379 nt, whereas introns without a G4 had a median length of 1629 nt (Supplementary Fig. 5a), (Mann–Whitney *U*, *p* value < 0.001). Interestingly, introns in the range of ~45–85 bps were the most enriched for G4s for both the 3′ss and the 5′ss. Moreover, the enrichment of introns in G4s declined rapidly with increased intron length, indicating that they are preferentially found in short introns (Fig. 3A, B, Kolmogorov–Smirnov test *p* value < 0.001). We also investigated the association between splicing strength score and intron length at sites with G4s in the

3′ss and 5′ss and found that the highest enrichment for G4s was in short introns with weak splice site strength (Fig. 3C). However, when comparing G4 enrichment at splice sites of short introns with a selection of long introns that have the same GC-content distribution, we found no difference or even higher enrichment of G4 at long intron splice sites (Fig. 3D and Supplementary Fig. 5b), indicating that the association of G4 to splice sites located at short introns could be driven by GC-content.

To further investigate the relationship regarding the intron length, we separated G4s identified using the consensus motif into non-template and template for both the 5′ss and the 3′ss. In this case, the GC-content is not a covariate since the template and non-template strands have the same GC-content. The 3′ss introns showed small but significant differences in length if a G4 was at the non-template or the template strand with medians of 736 nt and 621 nt, respectively (Mann–Whitney *U*, *p* value < 1e−16) (Supplementary Fig. 5c). However, if a G4 was at the non-template strand at the 5′ss the median intron length was 267 bp, whereas if the G4 was at the template strand the median intron length was 539 nt (Mann–Whitney *U*, *p* value < 1e−16)

(Supplementary Fig. 5c), displaying more aggravated differences in intron length. Therefore, we conclude that the highest enrichment is found for short introns, on the non-template strand, downstream of the 5′ss.

We also investigated if there is an association between G4s near splice sites and exon length. We do not find a significant association between G4s and exon length at the 3′ss (median exon length without G4s: 124 bp, median exon length with G4s: 123 bp, $p$ value >0.05, Mann–Whitney $U$), but we find a significant association for smaller exons near the 5′ss, albeit with a very small magnitude (median exon length without G4s: 127 bp, median exon length with G4: 123 bp, $p$ value <0.001, Mann–Whitney $U$). Furthermore, we explored if microexons, defined as exons <30 nt long[61–63] had an enrichment for G4s at their splice sites relative to other exons. However, we could not find a higher density of G4s at the introns flanking microexons than other exons.

In addition to exploring the relationship between intron and exon lengths and G4s, we also considered the position across the gene body. For each gene with nine or more exons, we separated the exons of its longest transcript into nine groups; the first four exons, the last four exons and the remaining middle exons. We find an attenuation of the enrichment around the last intron for the 5′ss and at the first couple of introns for the 3′ss, indicating clear differences at both ends of the transcript in comparison to other introns (Supplementary Fig. 5d), this result indicates that the role of G4s in splicing regulation is pervasive across the gene body.

Importantly, we also separately measured the G4 enrichment at the template and non-template strands of splice junctions across the gene length (Fig. 3E). We found that the enrichment of G4s was consistently higher in the non-template strand across exons. The foremost difference between enrichment scores for the two strands was found in the 3′ss exceeding 3-fold higher enrichment, while the differences between the two strands at the 5′ss were smaller. These findings provide evidence for widespread variation in the topography of G4s in splice junctions; these include the frequency of G4s in the exons and introns flanking the splice site, biases regarding the strand preference, the distance from the splice site and the positioning across the gene body.

**Dynamic splicing responses to stimuli are associated with proximal RNA G4s.** To gain further insights regarding the biological roles of G4s in the modulation of alternative splicing events, we investigated if G4s are associated with dynamic splicing changes in response to depolarisation stimuli. To that end we analysed published data from mouse and human embryonic stem cell (ESC)-derived neurons and mouse primary neurons subjected to a depolarisation solution including 170 mM of KCl and an L-type $Ca2^+$ channel agonist, resulting in an influx of $Ca2^+$ and followed by RNA-seq 4 h post-treatment[64]. The rise of intracellular $Ca2^+$ has been shown to have an impact on alternative splicing mediated by calmodulin-dependent kinase IV (CaMK IV)[65].

We used Whippet[66] to quantify alternative splicing events after depolarisation and compared them to the unstimulated controls (Methods) (Supplementary Material). The change in the inclusion of an exon is quantified using the percent spliced-in index (PSI) which represents the fraction of transcripts that contain the exon. We found a total of 44 G4-flanked exons that are differentially included in at least one human or mouse RNA-seq experiment. As case studies, we considered exons flanked by one or more G4s in the *SLC6A17*, *UNC13A* and *NAV2* loci that were differentially included after treatment for further experimental validation (Fig. 4A and Supplementary Fig. 6). Firstly, *SLC6A17* (NTT4/XT1) is a member of the SLC family of transporters that are involved in Na$^+$-dependent uptake of the majority of neurotransmitters at presynaptic neurons[67]. SLC6A17 is involved in the

transport of neutral amino acids, and mutations in this gene have been associated with autosomal-recessive intellectual disability[67,68]. We show that exon seven from *SLC6A17*, which is alternatively skipped after KCl treatment (Delta PSI = −0.177), has a G4 50 nt downstream of the 5′ss on the non-template strand. As the domains of SLC6A17 include an intracellular loop, two transmembrane regions and part of extracellular domains, the KCl-induced alternative skipping of this exon may lead to functional structural changes (Fig. 4A). Similarly, *UNC13A* encodes another presynaptic protein involved in glutamatergic transmission, and it has been associated with amyotrophic lateral sclerosis[69]. We identify a G4 downstream of exon 38, which results in dramatic exon skipping (Delta PSI = −0.369), (Supplementary Fig. 6a). Finally, the third target was a G4 located downstream of exon 16 in NAV2 (navigator protein 2), which is required for retinoic acid-induced neurite outgrowth in human neuroblastoma cells[70]. Again, KCl treatment resulted in exon skipping (Delta PSI = −0.271), which affects a NAV2 serine-rich sequence region (Supplementary Fig. 6b).

For each of the three candidates, we performed multiple assays to provide additional support for the formation of G4s at the RNA level[30,71,72]. First, we performed circular dichroism spectroscopy and UV-melting measurements of the G4-containing RNA oligonucleotides, in the presence of lithium ions (non-G4 stabilising) or potassium ions (G4 stabilising), to examine the formation potential and stability of RNA G4s. Supporting the case that RNA G4s are present in the transcripts, we found that all three candidates folded into stable RNA G4 structures preferentially under K$^+$ conditions (Fig. 4B–F and Supplementary Fig. 7a). To confirm the results from the circular dichroism and UV-melting experiments, we further used fluorescent-based arrays such as N-methyl mesoporphyrin IX (NMM) ligand enhanced fluorescence, Thiovlavin-T (ThT)-enhanced fluorescence, and intrinsic fluorescence experiments (Fig. 4G–I and Supplementary Fig. 7b, c). Indeed, we observed increased fluorescence intensity under conditions that promoted RNA G4 formation for all three candidates, confirming the formation of RNA G4s in these examples.

Having validated the formation of RNA G4s around three exons that are differentially included after KCl-induced depolarisation, we examined their impact on splicing genome-wide (Fig. 5A). We find that exon skipping at core exons is associated with G4s (chi-squared test multiple testing corrected, $p$ value <5e−12, Fig. 5A). We also found G4s associated with the inclusion of alternative first exons following KCl treatment indicating alternative promoter usage (chi-squared test, multiple testing corrected, $p$ value <0.05, Fig. 5A). The analysis identified a total of 22,344 splicing events where the absolute value of Delta PSI was greater or equal to 0.1 and the probability was greater or equal to 0.9. We focused our analysis on the 2633 events that involved cassette exons, and of these 2346 (89.1%) corresponded to increased skipping (Fig. 5B, binomial test, $p$ value < 0.001). These results are consistent with previous studies which have demonstrated exon skipping following depolarisation in individual examples[73–77] and genome-wide analyses of RNA-seq experiments[78]. Interestingly, we found enrichment for differential splicing events with G4 motifs at the associated splicing sites, exceeding that which was expected by the background distribution (Fig. 5A–C, chi-squared test with multiple testing correction, $p$ value <0.001, odds ratio = 1.57). To provide further support for the findings obtained from the consensus G4 motif, we examined the distribution of G4-seq derived peaks in PDS and K$^+$ conditions around splice sites of differentially and non-differentially included exons. As expected, we found a consistent enrichment at the differentially included exons (Fig. 5A and Supplementary Fig. 8). Moreover, the effect size was larger for the

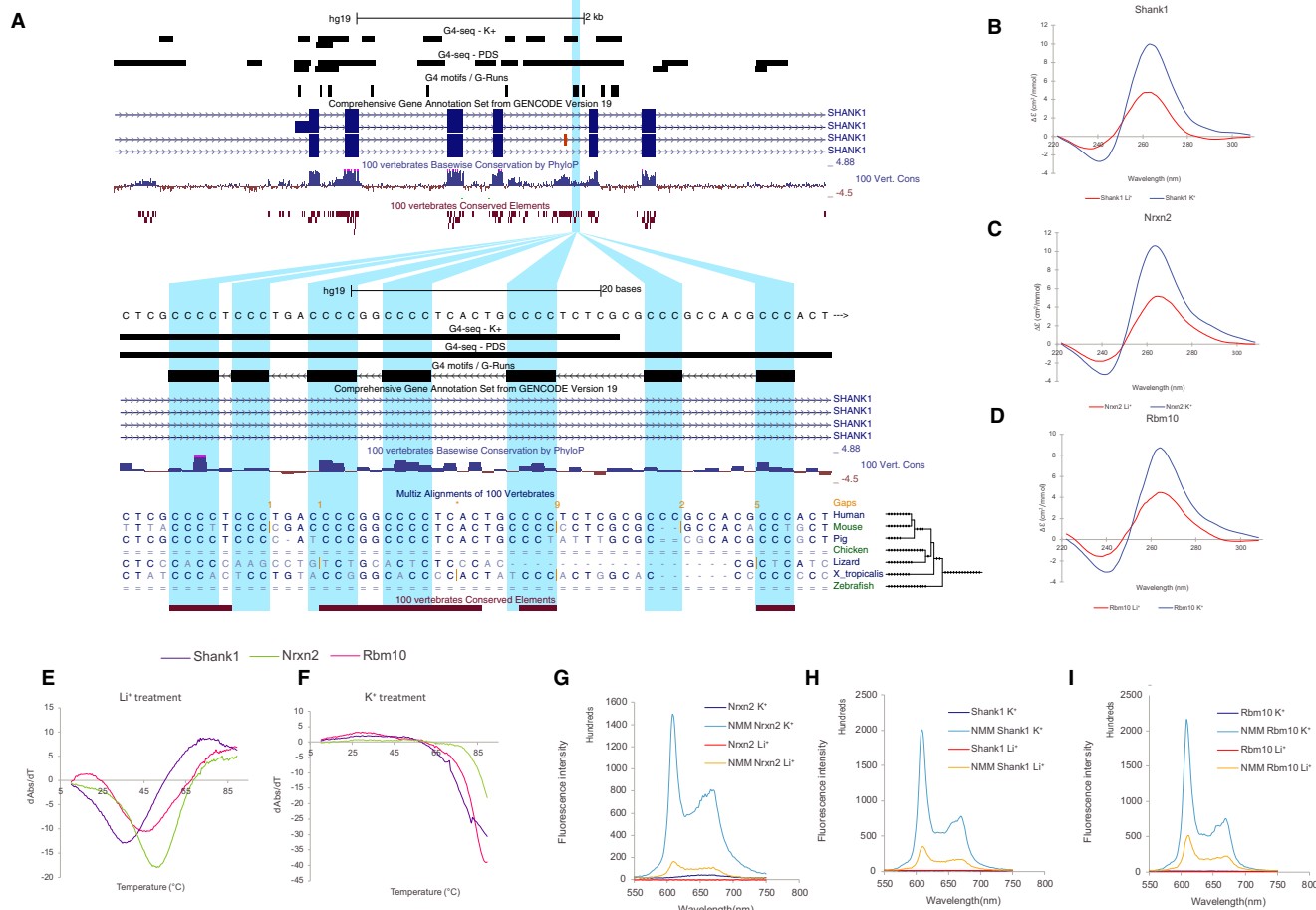

**Fig. 4 Targeted validation of RNA G4s found in splicing sites in presence of a G4 stabilising cation (K⁺) and a non-G4 stabilising cation (Li⁺). A** UCSC screenshot displaying the *SLC6A17* locus and the distribution of G4 consensus motifs, G4-seq derived G4 peaks and protein domains. **B–D** Circular dichroism (CD) spectra of the three candidate targets for RNA G4 formation potential in presence of two cations. The monovalent ion-dependent nature (G4 stabilised in K⁺ but not in Li⁺) and the CD profile (positive peak at 262 nm and negative peak at 240 nm) indicate the formation of RNA G4s with parallel topology. **E, F** UV-melting profiles of the three G4 candidates in presence of Li⁺ and K⁺. Hyperchromic shift at 295 nm is a hallmark for G4 formation, which can be transformed to a negative peak in the derivative plot (dAbs/dT) for G4 stability analysis. The melting temperature (Tm) of a G4 can be identified at the maximum negative value. For the K⁺ treatment, the Tms of the RNA G4 are >85 °C. **G–I** Fluorescence emission associated with NMM ligand binding to RNA G4 candidates in the presence of Li⁺ or K⁺ ions. In the absence of NMM ligand, no fluorescence was observed at ~610 nm. Upon NMM addition, weak fluorescence was observed under Li⁺, which was substantially enhanced when substituted with K⁺, supporting the formation of RNA G4 which allows recognition of NMM and enhances its fluorescence.

G4 motifs and the G4-seq derived DNA G4 sites in the non-template strand at the 5′ss in comparison to those found at the template strand (chi-squared test multiple testing correction, *p* value <0.001 when using the consensus G4 motif and for both PDS and K⁺ G4-seq conditions in human neurons). In addition to human cells, we performed the same analysis in similar experiments in mice across four different conditions (Supplementary Figs. 8–11). We recapitulate the widespread exon skipping phenomenon observed after depolarisation in human neuronal cells (Fig. 5A and Supplementary Fig. 9), but we found a significant association of alternative included exons only with PDS G4-seq derived peaks in two conditions (Supplementary Fig. 11). Importantly, we report alternative promoter usage associated with G4s and alternative first exon inclusion in both mouse and human neurons following KCl treatment (Fig. 5A and Supplementary Fig. 10). Taken together, our results suggest that the presence of G4s at the splice junction of cassette exons is associated with dynamic changes of alternative splicing in response to KCl-induced depolarisation.

**Mutations at G4s affect splicing.** To provide direct evidence for the role of G4s in the modulation of alternative splicing events, we designed two minigene constructs that contain wild-type and mutant G4 motifs, which we previously validated to lead to RNA-G4 structure formation in the *SLC6A17* and *NAV2* genes (Fig. 4). Within these constructs, we included the whole sequence of exons that were differentially included after KCl-induced depolarisation and corresponding flanking regions (Supplementary Fig. 12). In the case of *SLC6A17* minigene, we inserted a wild-type or mutated G4 motif flanked by two exons, one of them corresponding to a KCl-responding alternative exon (Fig. 5D). In the *SLC6A17* minigene containing the wild-type G4 motif, we observed two main splicing products corresponding to isoforms, where either both exons are included or excluded and a minority product where only one alternative exon is included (Fig. 5E). After the introduction of mutations in the G4 motif, we observed strong exclusion of both exons. Similarly, for the *NAV2* minigene experiments, we also observed that mutations over the flanking G4 motifs favoured exon skipping (Supplementary Fig. 12). These

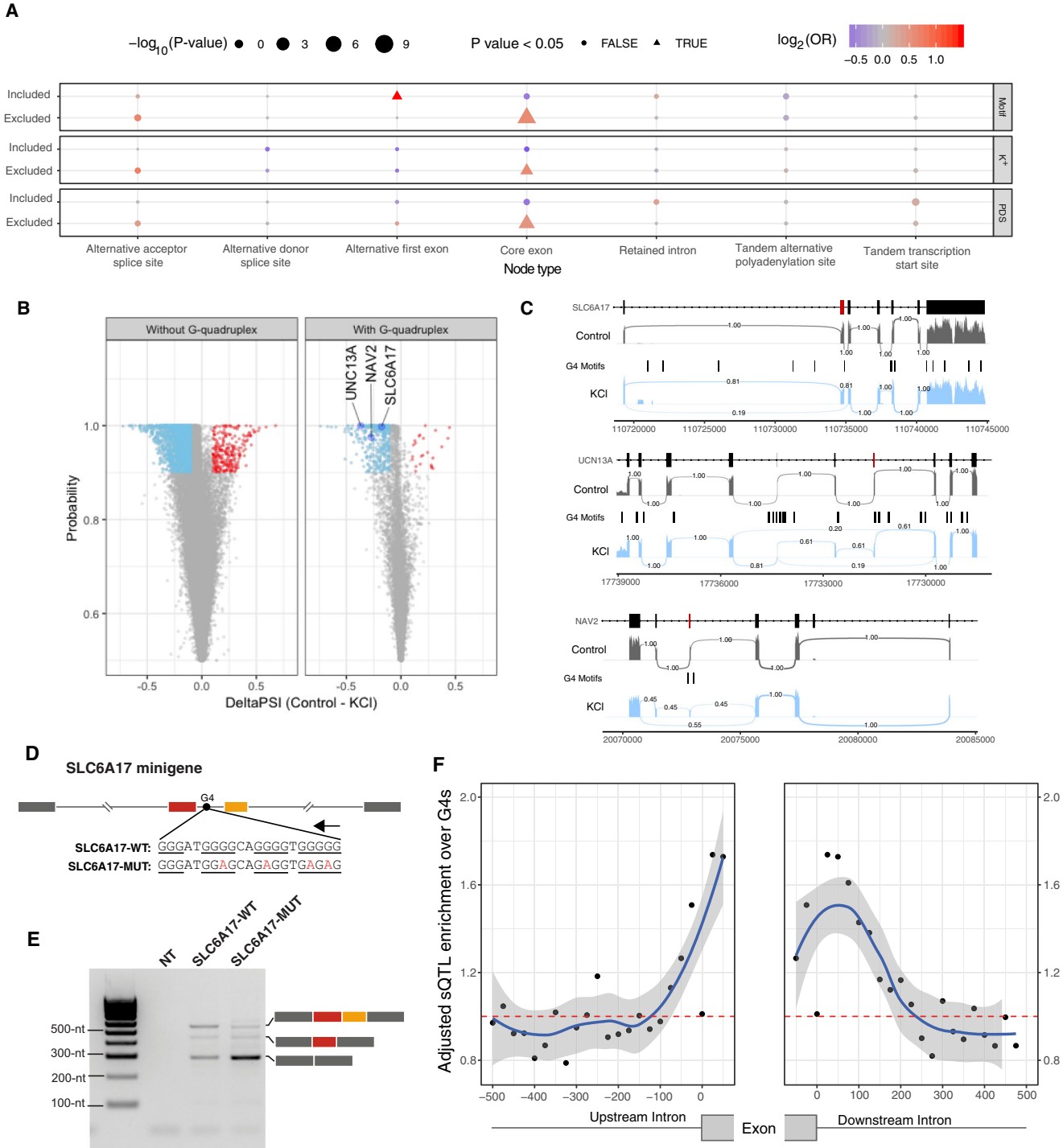

results indicate that G4 motifs present in flanking intronic regions can have a direct effect on alternative splicing outcomes, favouring exon inclusion, which is in agreement with previous observations[39].

To investigate if G4 motifs have a transcriptome-wide effect over alternative splicing, we took advantage of splicing quantitative trait loci (sQTL) data from GTEx consortium data[79]. Since DNA G4s are reported to be enriched in both germline and somatic mutations[28,80], we adjusted the enrichment of sQTLs for differences in SNP distribution across the splice junction. We found that sQTLs were more likely to overlap G4 motifs that are in close proximity to splice sites (Fig. 5F). The highest sQTL adjusted enrichment values were found in exonic regions and the most proximate flanking intronic regions. Therefore, we conclude that G4 motifs are enriched for sQTLs relative to the expected population variant frequency, suggesting a functional role.

**Systematic characterisation of the interplay between RBPs and G4s at splice sites.** During gene expression, RNA-binding proteins (RBPs) enable different catalytic steps of RNA processing and serve as key regulatory factors of alternative splicing. We processed data from 1345 eCLIP experiments that were performed on K562 and HepG2 cell lines[81] to calculate the differential binding of RBPs between exons that are flanked and exons that are not flanked by G4 motifs within a 200 bp window. We

**Fig. 5 Neuronal stimulation with KCl results in alternative splicing events associated with the presence of G4s. A** G4 association with alternative splicing changes. Inclusion and exclusion patterns of splice nodes are shown in association with DNA G4 presence or absence following KCl treatment. The odds ratio represents the relationship between the presence of G4s and alternative splicing changes. The odds ratio significance was assessed by chi-squared tests. All p values were calculated with chi-squared tests using Yates' correction and also adjusting for multiple testing with Bonferroni corrections. **B** Volcano plot showing differential inclusion events in the presence and absence of flanking G4 motifs and the associated probability following potassium stimulation with widespread exon skipping after depolarisation in human neuronal cells. **C** Sashimi plots showing alternative exon inclusion for the three candidates, namely SLC6A17, UNC13A and NAV2 following KCl treatment. Exons flanked by an RNA G4 that were used for validation experiments (Fig. 4 and Supplementary Fig. 6) are marked in red. The numbers connecting exons represent the fraction of reads supporting each path. **D** Schematic showing the design of a minigene assay to test the effect of G4 motifs on alternative splicing. Two consecutive exons from *SLC6A17* and their flanking intronic regions were inserted into the minigene construct. The red exon corresponds to the alternative exon highlighted in **C**, while the orange exon represents the following downstream exon that was not detected as alternatively included after KCl treatment. The arrow above the G4 motif indicates its location on the non-template strand. Wild-type and mutant G4 sequences are shown with G-runs underlined and mutations in red. **E** Minigene assay results show the effect of mutations in G4 motifs, which strongly promote the skipping of both SLC6A17 exons. This experiment was conducted twice, leading to reproducible results (Supplementary Fig. 12e). **F** Adjusted enrichment of sQTLs at G4s relative to splice sites. The error bands represent 95% confidence intervals based on a binomial model.

performed unsupervised hierarchical clustering of RBPs based on their G4 motif enrichment profile, taking into consideration G4 motifs at the template and non-template orientations in proximity to splice sites. Using unsupervised hierarchical clustering we identified a total of ten clusters with distinct differential binding patterns (Fig. 6A), of which all except cluster 6 showed substantial and significant enrichment differences between exons that are flanked by G4 motifs and exons that are not flanked by G4 motifs. However, the other constituent clusters exhibited clearly distinguishable patterns; for instance, in cluster 7 we observed an enrichment difference only for G4s found at the non-template orientation, whereas for cluster 10 we observed an enrichment difference specific to the template orientation. These results show that several RBPs differentially bind to splice sites flanked by G4 motifs. To complement our observations from eCLIP data, we analysed 506 loss of function (LoF) experiments followed by RNA-seq that targeted a total of 269 RBPs in HepG2 and K562 cell lines, from which 143 RBP factors overlapped between the eCLIP and LoF experiments. We performed quantitative analyses to determine the number of differentially included exons induced by the LoF of target RBPs. We found that for 36 RBPs, differential inclusion following the LoF of the RBP is associated with the presence of a G4 motif in proximity to the splice site in at least one of the analysed LoF experiments (chi-squared test, adjusted p value < 0.05) (Fig. 6B). Integrating eCLIP with LoF RNA-seq experiments we obtained a high-confidence list of 15 differentially bound RBPs to G4 motifs that show a significant and consistent association with alternative splicing (Fig. 6B). Interestingly, we found examples such as HNRNPK and HNRNPU, which exhibit higher binding to G4 motif around splice sites and are positively associated with differentially included exons (p value < 0.05 and $\log_2(OR) > 0$), suggesting that these factors could have a direct impact on G4-mediated AS regulation (Fig. 6C). Conversely, we also found examples such as AQR and RBM15, which are depleted around splice sites flanked by G4 motifs and are negatively associated with differentially included exons (p value < 0.05 and $\log_2(OR) < 0$), suggesting that binding and impact of these factors over AS could be prevented by G4 formation (Fig. 6C). We also found cases such as RBFOX2, that exhibit remarkably different binding profiles across splice sites flanked by G4 motifs, although they were not positively or negatively associated with differentially included exons.

**Enrichment of DNA G4s at splice sites does not extend beyond vertebrates.** Alternative splicing is a pivotal step of eukaryotic mRNA processing. To understand to what extent splice site regulation by G4s is conserved we considered eleven eukaryotes: *Homo sapiens* (human), *Mus musculus* (mouse), *Sus scrofa* (pig),

*Gallus gallus* (chicken), *Danio rerio* (zebrafish), *Caenorhabditis elegans* (nematode) *D. melanogaster* (fruit fly), *Xenopus tropicalis* (frog), *Anolis carolinensis* (lizard), *Saccharomyces cerevisiae* (yeast) and *Arabidopsis thaliana* (flowering plant). *S. cerevisiae* was excluded from further analysis since we could not find any DNA G4s at splice sites and DNA G4s were rare with only 39 occurrences genome-wide. Interestingly, we found that the enrichment pattern of DNA G4s at splice sites was restricted to a subset of vertebrate species, with minimal or no enrichment in fruit fly, *Arabidopsis* and *C. elegans* (Fig. 7A, B and Supplementary Fig. 13a). We observed strong enrichment in chicken, pig, human and mouse, while lizard displayed limited enrichment levels. Surprisingly, frog and zebrafish displayed relative depletion. This suggests that alternative splicing regulation by G4s is found to be restricted to mammals and birds, but absent in plants, other tetrapods or fish.

Additional support for this conclusion comes from our analysis of G4-seq derived DNA G4 maps generated in PDS and $K^+$ conditions. These maps are available for multiple model organisms, including three vertebrates (human, mouse and zebrafish) and four non-vertebrate species (nematode, fruit fly, arabidopsis and yeast). Consistent with the analysis based on the primary sequence, we find an acute enrichment of DNA G4s at the 5'ss and 3'ss only in human and mouse. In particular, we could not find any DNA G4s in the vicinity of splicing junctions for *S. cerevisiae*, there was no enrichment for *D. melanogaster* and *D. rerio*, while we observed a depletion in *A. thaliana* (Fig. 7C, D).

## Discussion

The identification of splicing regulators remains an active area of research, as the information content at splice sites is insufficient for predicting alternative splicing events in higher eukaryotes[8,82]. Here, we provide evidence for the widespread role of G4s in splicing regulation. The enrichment of DNA G4s at splice junctions is comparable to what is observed[10] at promoters in humans (Supplementary Fig. 13b), even though it is primarily the importance of G4s for transcriptional and translational regulation that has previously been recognised[48,83]. We provide several lines of evidence, including a high enrichment at splice site regions, preference for the non-template strand and in vitro experiments to suggest that G4s form at the pre-mRNA and can modulate alternative splicing events. In addition, RNA G4s are more stable than DNA G4s, suggesting that they could have a greater influence in the transcriptome than in the genome[83,84]. However, co-transcriptional splicing has been previously demonstrated to be the norm[85,86], and we cannot rule out the possibility that the nascent transcripts form DNA-RNA hybrids, implying more complex interactions[87]. In fact, enrichment of G4s at the template

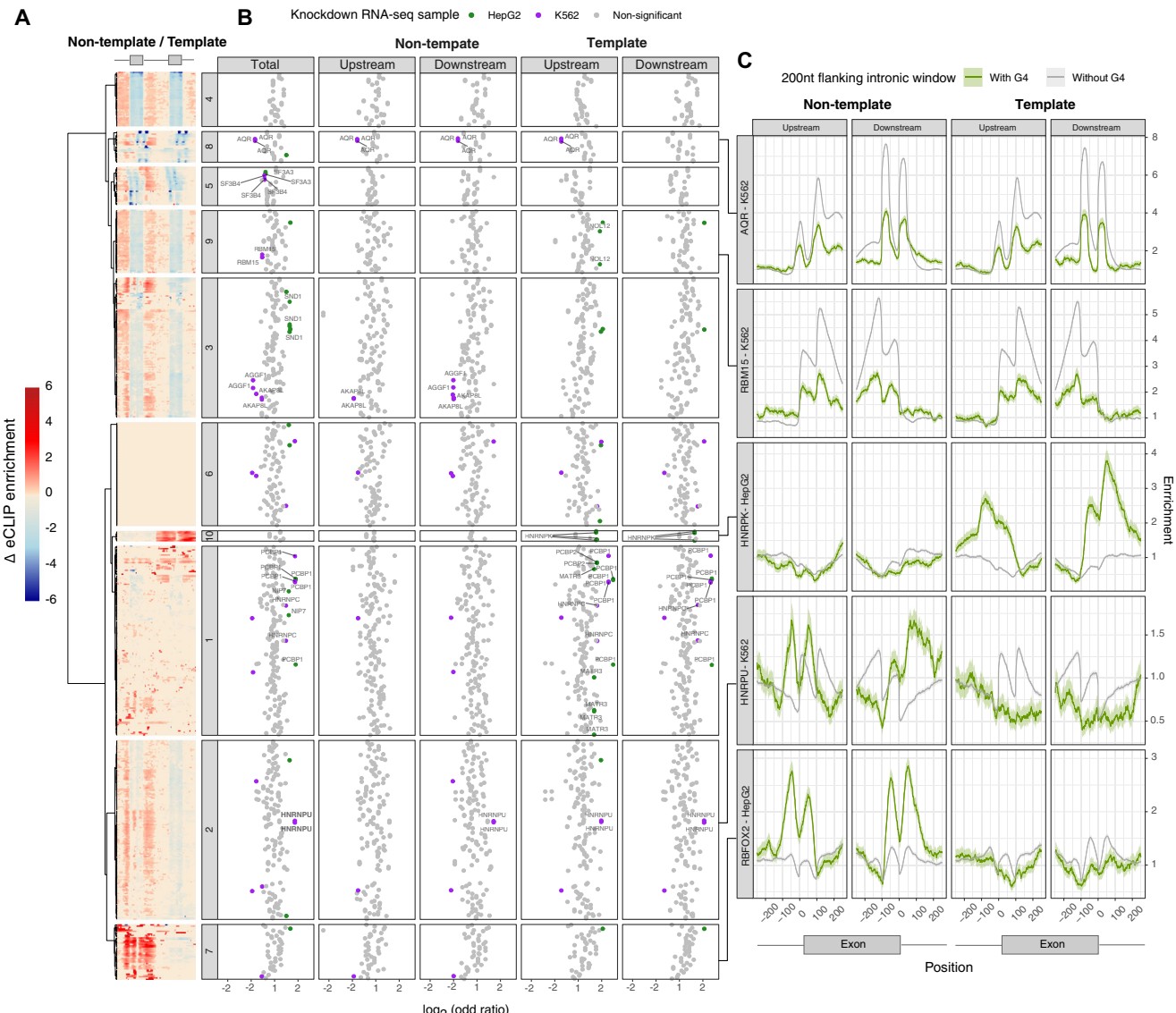

**Fig. 6 The interplay between RNA binding proteins and G4 motifs in proximity to splice sites. A** Enrichment difference in the distribution of RBPs in sites with and without G4 motifs is shown for the template and non-template G4 orientations. Heatmap displaying the clustering of differential enrichment values of each RBP between splice sites flanked by non-template (left) or template (right) G4s. **B** Functional assessment of the RBP effect on alternative inclusion of exons flanked by G4 motifs. Each dot represents a single replicate performed for an RBP on a given cell line. Log(odds ratio) of differential inclusion for sites with and without G4 motifs, following RBP KO experiments. Significant associations between RBPs and alternative inclusion of G4-flanked exons are highlighted either in green (HepG2) or purple (K562). Labelled factors are the set of high-confidence factors identified by consistent eCLIP and LoF followed by RNA-seq analyses. Results are shown in aggregate, as well as for template and non-template orientations separately. Statistical evaluation was performed using chi-squared tests with Bonferroni corrections. Results for both eCLIP and RNA-seq experiments that target the same factor are shown side by side. **C** Enrichment of a panel of RBP binding sites across splice sites flanked or not flanked by G4 motifs. Displayed factors are AQR, RBM15, HNRPK, HNRPU and RBFOX2, each of which belongs to a different eCLIP cluster. The error bands represent 95% confidence intervals based on a binomial model.

strand suggests formation and potential roles at the DNA level as well, likely during co-transcriptional splicing when the DNA is single-stranded or the formation of i-motifs[88].

Weaker splice sites lead to suboptimal exon recognition, which enables alternative splicing to be modulated by additional cis-regulatory elements or epigenetic factors[89]. Here we show a pronounced enrichment of G4s at weak splice sites and provide evidence for a widespread contribution of G4 structures over alternative splicing. We also find that G4s appear in a subset of species near splice sites (Fig. 7), suggesting that they have emerged as splicing modulators during vertebrate evolution. The presence of additional regulatory mechanisms is in accordance

with higher frequencies of alternative splicing events in vertebrates compared to invertebrates[90]. Moreover, DNA G4s display a higher likelihood of DNA mutations[91] and as a result they are likely plastic in nature, enabling rapid splicing changes during evolution and the establishment of new functions through alternative splicing and the generation of isoform diversity.

We observed widespread exon skipping following potassium depolarisation of neurons (Fig. 5A and Supplementary Fig. 9), a phenomenon that, to our knowledge, has only been documented for a handful of cases[73,74,76,77]. These alternative splicing changes are likely induced by $Ca^{2+}$ influx after depolarisation which is known to affect splicing via CaMK IV[65]. Here, we show that the

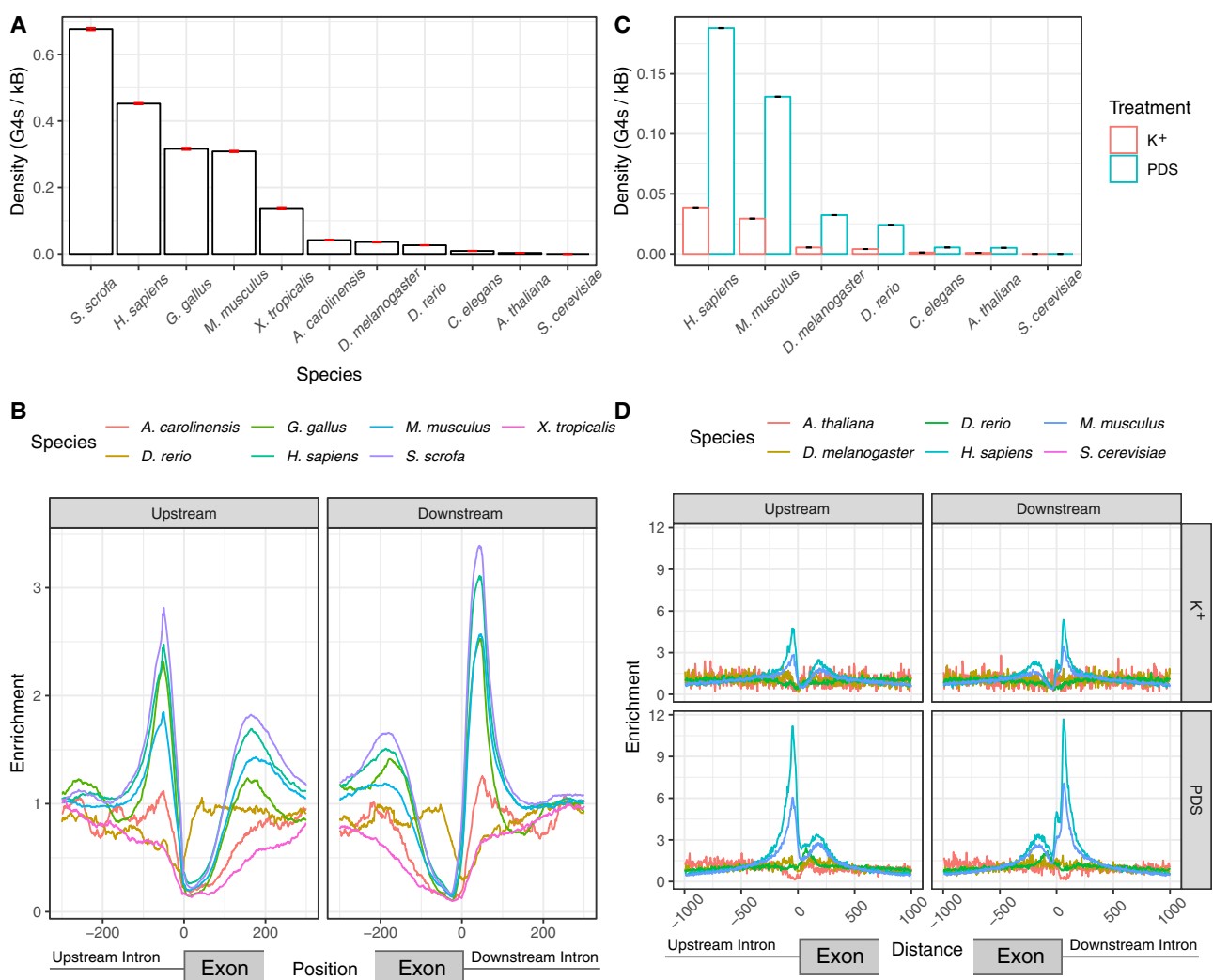

**Fig. 7 Enrichment of DNA G4s at splicing sites across different species. A** Density of G4 motifs in a 100 nt window on each side across all upstream (3'ss) and downstream (5'ss) splice sites of each species, measured as mean values. Error bars indicate standard deviation from 1000-fold bootstrapping with replacement. **B** Enrichment of G4 motifs at splice sites for seven vertebrate species, using the consensus G4 motif. **C** Enrichment of G4-seq derived DNA G4s at splicing sites at 100 nt splicing site windows in PDS and K$^+$ treatments, measured as mean values. Error bars indicate standard deviation from 1000-fold bootstrapping with replacement. **D** Enrichment of DNA G4s at: upstream (3'ss) and downstream (5'ss) splice sites across six species for PDS and K$^+$ treatments.

changes in splicing patterns are associated with the presence of G4s at the splice junctions. Given the relatively short interval between the time points for the RNAseq samples, we find it most likely that the changes in splicing are due to post-translational effects rather than altered expression of splicing regulators. In fact, part of the alternative splicing changes observed in response to depolarisation has been shown to be dependent on hnRNP L phosphorylation by CaMK IV[76].

Our results also show that G4 motifs present at flanking intronic regions can have a direct over alternative splicing, evidenced by sQTL enrichment analyses at G4s nearby splice sites and by our minigene experiments (Fig. 5 and Supplementary Fig. 12). These results corroborate the importance of G4s as splicing modulators, and our findings are consistent with previous work[39]. We also provide evidence that some RBPs' binding preference to splice sites depends on the presence of G4s (Fig. 6). We find 15 RBPs whose binding and expression perturbation profiles are significantly associated with the presence of G4s at splice sites, such as HNRPU, HNRPK, RBM15 and PCBP2, suggesting that they could be involved in the mechanism by

which G4 formation modulates alternative splicing (Fig. 6). However, it remains unclear by which mechanism G4s influence alternative splicing. G4 formation has the potential to mask otherwise accessible RBP sites, which could explain RBP binding profiles of factors such as AQR, SF3B4, SF3A3 and RBM15, which we found to be depleted in G4-flanked splice sites, and their negative association with alternative splicing changes across the LoF RNA-seq experiments. On the other hand, experimental evidence suggests that G4 stabilisation favours the binding of proteins such HNRNP H/F[39,92] and HNRPU[93]. Our results show that HNRPU has stronger binding profiles across G4-flanked exons, which are also enriched amongst detected alternative splicing events after HNRPU knockdown, supporting the observations from Izumi and Funa[93]. Moreover, similar results were observed for HNRPC, MATR3 and PCBP1/2, suggesting additional RBPs that can mediate the effect of RNA G4s over alternative splicing. Finally, we also found significant differential RBP binding profiles for other factors such as DDX3X, PRPF4, GTF2F1 and CSTF2T (Supplementary Fig. 14) which were previously linked to RNA G4s[94,95] but were not highlighted by our

RNA-seq analyses, suggesting the possibility that LoF effects of some of these factors could not be detected due to the strict criteria that we implemented to assess the consistency across all experiments and cell lines.

The fact that G4 formation at the RNA level can be modulated by helicases, monovalent ions or small molecules[41,96–98], opens up new avenues for modulating splicing for therapeutic purposes. It is plausible that by perturbing the stability of RNA G4s or the lifetime of the folded state other RBP binding sites can either become exposed or masked, modulating alternative splicing events. Drugs targeting splicing modulation are already clinically approved, such as Spinraza for spinal muscular atrophy[99]. There are already multiple compounds available with varying specificities for G4 binding. One example is Quarfloxin, which previously reached phase II clinical trials targeting a DNA G4 in the *CMYC* promoter, but its evaluation was halted due to interference with pol I in rRNA[100]. Our work suggests that G4s in splice sites could be used as pharmacological targets.

## Methods

**Genome and gene annotations processing**. We obtained genome assemblies from the UCSC Genome Browser FTP server for eleven organisms: GRCh37 (hg19) reference assembly of the human genome, the mouse reference genome (mm10), the *Saccharomyces cerevisiae* reference genome (sacCer3), the chicken reference genome (galGal5), the *Drosophila melanogaster* reference genome (dm6), the zebrafish reference genome (danRer11), the *Xenopus tropicalis* reference genome (xenTro9), the *Anolis carolinensis* reference genome (anoCar2), the *Arabidopsis thaliana* reference genome (Tair10) and the *Caenorhabditis elegans* reference genome (ce10).

We downloaded the Ensembl gene annotation files for the associated genomes from UCSC Table Browser as BED files for each species[101]. Using in-house python scripts we extracted the coordinates of internal exons flanked by canonical splice sites (GT-AG introns) for every species. To calculate the splicing strength scores, we used publicly available positional frequency matrices from the SpliceRack database[58] and previously developed scripts used before for the same purpose[55]. Splice sites were grouped into quartiles based on their splicing strength score for the downstream analyses to study the distribution of non-B DNA motifs and in particular G4 motifs (Fig. 2A and Supplementary Figs. 3 and 4). For Fig. 2A, the confidence intervals were calculated using "binconf" command from the "Hmisc" package in R with default parameters. Mann–Whitney U tests were performed at 100 nt on each side in the upstream splice site and at the downstream splice site to compare the splicing strength scores of sites with and without G4s.

We used in-house scripts to process a bed file containing all annotated alternative events which were obtained from VastDB's UCSC Genome Browser Track Hub[53]. We extracted the splice site coordinates from exon skipping (HsaEX), alternative acceptor (HsaALTA), alternative donor (HsaALTD), intron retention (HsaINT) and mutually coordinated events (MULTI) to analyse each category separately for G4 enrichment.

### Genomic datasets

*Non-B DNA motifs*. Identification of each non-B DNA motif was performed using the genome-wide maps in humans and mice provided by[102] and processed as described in[28]. We focused on seven non-B DNA motifs; inverted repeats, mirror repeats, H-DNA which forms at a subset of mirror repeats with high AG content, G4s, Z-DNA which forms at non-AT alternating purine pyrimidine stretches, short tandem repeats and direct repeats (Fig. 1A and Supplementary Fig. 3).

Regular expressions were employed to identify genome-wide consecutive G-runs across the human genome, interspersed with loops of up to 7 bps. In total, one to six consecutive G-runs were searched (Fig. 2B). For each species, we generated the genome-wide DNA G4 maps using a regular expression of the consensus G4 motif (G ≥ 3N1-7G ≥ 3N1-7G ≥ 3N1-7G ≥ 3) (Fig. 7A, B). Orientation of G4s and G-runs was performed with respect to the template and non-template strands to calculate strand asymmetries at genic regions as previously described for polyN motifs (N being Gs, Cs, Ts and As) in[103,104] (Fig. 2A and Supplementary Figs. 2 and 4e–h).

Permuted windows of 100 nt on each side of each splice junction were generated using ushuffle[105] correcting for dinucleotide content. The fold enrichment for G4s was calculated as the ratio of the number of motifs found in the real sequences and the median of 1000 permutations of the set of all real sequences. The corrected enrichment of G4s at 3′ss and 5′ss was calculated as the ratio of the real enrichment of G4s over the background enrichment of G4s at shuffled splice site windows.

To investigate the relationship between non-B DNA motifs or G4-seq peaks and splice sites, we generated local windows around the splice sites and measured the distribution of each non-B DNA motif (Fig. 1A, D and Supplementary Fig. 3) or for G4-seq peak base pairs across the window (Fig. 1B and Supplementary Fig. 1a, c).

The enrichment was calculated as the number of occurrences at a position over the median number of occurrences across the window. Regardless of the window size shown in the figures, the enrichment was calculated over a window of 1kB. The same approach was used to calculate the enrichment of G4s at splice sites across different species (Fig. 7B).

The density of G4 consensus motifs or G4-seq derived peaks at local windows was calculated as the number of occurrences of the motif or the peak over the total number of base pairs examined (Fig. 7A, C).

*G4-seq data*. G4-seq BedGraph data were obtained from GEO accession code GSE63874[50] for the human genome and analysed with bedtools closest command to identify the closest DNA G4 to splice sites and calculate the distance. The analysis was performed separately for Na⁺-K⁺ and Na⁺-PDS conditions and it was compared to the distribution obtained from the G4 consensus motif. G4-seq BedGraph data for six species, human, mouse, *D. melanogaster, C. elegans, A. thaliana* and yeast, were obtained from GEO accession code GSE110582[51] and analysed using the same genome annotations as those used for the generation of each G4-seq dataset.

Coordinates for internal exons flanked by canonical splice sites (GT-AG introns) were extracted for each species using the Ensembl annotation versions described in[51] using custom Python scripts.

*Relationship between G4s and exon/intron length*. Introns and exons were grouped based on the presence or absence of G4s within 100 nt on each side of the 5′ss and 3′ss and further subdivided into those containing a G4 on the template or on the non-template strand, separately for the 3′ss and the 5′ss. For each of the eight groups, we calculated the median length of the intron or exon in a group and performed Mann–Whitney U tests to calculate the significance of the association between the length of exons/introns and G4 presence (Supplementary Fig. 5a). The R function stat_density was used to plot the length distribution of introns with and without G4s as modelled by a kernel density estimate (Fig. 3A). Abundance enrichment of intron length in 3′ss/5′ss in relationship with the presence of G4s was generated in R using the function geom_smooth in an eighth-grade model (Fig. 3B). Correction of GC content in introns with different lengths was performed by grouping introns into small introns (<500nt) and large introns (>500nt). Then we calculated the GC content for both groups and for each short intron we selected a long intron with a close GC content value, in such a way that GC distribution across short and long introns groups were nearly identical (Supplementary Fig. 5b).

*G4s and relationship to exon number*. For the longest transcript of each gene with nine or more exons, we separated exons into 9 groups, the first four exons, the last four exons and the remaining middle exons. To compare the frequency of G4s in splice junctions across the gene body we calculated the distribution of G4s in each exon group relative to the 5′ss/3′ss (Supplementary Fig. 5d). We also calculated the distribution of G4s in each exon group relative to the 5′ss/3′ss separately for the template and non-template strands as described previously[104] (Fig. 3E).

*Relationship between G4s, splicing strength score and intron length*. We calculated the splicing strength score and intron length for the upstream and downstream intron of each exon. We separated introns and splicing strength scores into deciles and calculated the G4 density at each decile, from which we produced two heatmaps displaying the density of G4s as a function of splicing strength score and intron length for the upstream and downstream introns (Fig. 3C).

### Comparative analysis of RNA-seq experiment

*Differential exon inclusion following depolarisation*. We analysed available data (BioProject Accession: PRJEB19451, ENA link: ERP021488) for mouse and human ESC-derived cortical neurons, mouse primary cortical neurons from wild-type and Tc1 mice stimulated with KCl treatment and untreated followed by RNA-seq at 4 h post-treatment[64]. These corresponded to two experiments in human cells and 12 in mouse cells. We used Whippet[66] to quantify splicing nodes and assess their alternative inclusion after KCl treatment and controls (Fig. 5A, B and Supplementary Figs. 9 and 10). We used the absolute value of Delta PSI greater than 0.1 and probability greater than 0.9 to define a splicing node as differentially included between treatments and controls.

We calculated the distance between the middle point of G4 motifs and G4-seq peaks from each splicing node to determine their association with G4s. Splicing nodes whose splice sites were within 100 bps to G4 motif or 45 bps to G4-seq peak were classified as G4 associated splicing nodes. Next, we assessed the influence of G4s on splicing changes following KCl depolarisation of human and mouse neurons by calculating the odds ratio score of each splicing node type. To determine the statistical significance of the effect we performed a chi-squared test using Yates' correction and also adjusting for multiple testing with Bonferroni corrections (Fig. 5A and Supplementary Fig. 10). The distribution of G4 motifs and G4-seq peaks was profiled around differentially included and non-differentially included core exon splicing nodes (CE) (Supplementary Figs. 8 and 11). The confidence intervals were calculated using "binconf" command from "Hmisc" package in R with default parameters. Sashimi plots were generated using "ggsashimi" package[106]. Inclusion and exclusion path ratios were calculated using

the total amount of spliced reads supporting each splice junction, where inclusion paths were calculated using the average read count for splice junctions flanking each exon side.

The differential gene expression analysis was performed using a Snakemake pipeline[107] that was adapted from a publicly available repository[108]. Reads were mapped with HISAT2 (v2.1.0)[109] and later quantified using FeatureCounts[110]. Finally, differentially expressed genes were identified using DEseq2 (v1.18.1)[111], based on the adjusted *p* values.

Three putative non-template G4s found in proximity to splicing junctions and which were differentially included following depolarisation in human ESC-derived neurons and at least one condition in mice were selected for validation experiments. These were: (1) a G4 downstream of exon 7 for *SLC6A17* (chr1: 110734886-110734906), (2) a G4 downstream of exon 38 in *Unc13a* (chr19:17731307-17731346) and (3) a G4 upstream of exon 16 in *Nav2* (chr11:20072958-20072979) for which RNA oligonucleotides at the G4 locations were ordered.

The RNA oligonucleotides used were (G-runs marked in bold):

(1) SLC6A17 oligonucleotide:
    **GGG**AGT**GGG**CA**GGGG**T**GGGGG**
(2) UNC13A oligonucleotide:
    **GGGGGG**TGGT**GGG**T**GGGGGGG**TTGGT**GGG**TA**GGG**CAGA**GGG**
(3) Nrxn2 oligonucleotide:
    **GGGGG**TTT**GGG**CT**GGG**CT**GGGG**

### Integration of eCLIP and LoF followed by RNA-seq experiments

*eCLIP data analysis*. eCLIP data for K562 and HepG2 cell lines were derived from the ENCODE consortium[81], including 1346 experiments of 150 RBPs. Among them, 722 experiments were performed in the K562 cell line and the remaining 624 experiments were performed in the HepG2 cell line, for 120 and 103 RBPs, respectively. To investigate the relationship between G4 sites, we extracted splice sites flanked by G4 within 100-bp intronic windows, with either template or non-template orientation. Splice site regions were then separated into 10 bp bins. For each bin, we calculated the factor binding enrichment across the different groups of splice sites (flanked by template or non-template G4s, or not flanked by G4s), and we then calculated the differential enrichment values between splice sites flanked and not flanked by G4s. We assessed the statistical significance of these differences using chi-squared testing with Bonferroni correction. For clustering based on differential enrichment, non-significant differences were set to 0. Differential enrichment values of template and non-template G4 bins were clustered using Ward's method with unsupervised hierarchical clustering with the hclust package in R to classify RBP enrichment profiles into ten clusters.

*Analysis of LoF experiments followed by RNA-seq*. We analysed 506 LoF experiments followed by RNA-seq that targeted a total of 269 RBPs in HepG2 and K562 cell lines, all of which were derived from the ENCODE consortium[81]. To quantify alternative splicing changes after knockout or knockdown of target RBPs, we used Whippet[66], which led us to obtain sets of differentially included exons associated with each LoF condition. Similarly to the KCl-induced depolarisation experiment analyses, we used the absolute value of Delta PSI >0.1 and probability >0.9 to define a splicing node as differentially included between treatments and controls. For each LoF experiment, we calculated the association between G4s and differentially included exons estimated as log odds ratio and assessed the statistical significance using chi-squared test with Bonferroni correction. To get a list of high-confidence factors that have a role in G4-mediated alternative splicing, we considered factors that were found to have significant eCLIP and RNA-seq analyses performed using the same cell line (K562 or HepG2). Finally, to check for experimental consistency, only factors with at least two eCLIP replicates that clustered in the same group were considered to be high confidence and they were labelled in Fig. 6B.

### Evaluation of sQTL data and G4s

*sQTL data*. sQTL data were derived from the GTEx consortium[79] from the link https://storage.googleapis.com/gtex_analysis_v8/multi_tissue_qtl_data/GTEx_Analysis_v8.metasoft.txt.gz. Population variants were derived from dbSNP153. The analysis was done within 25 bp windows from −500 bp to +500 bp relative to the splice site. The enrichment for sQTLs and SNPs was calculated as the density of their occurrences within G4s in each bin relative to the density across all bins. The adjusted enrichment for sQTLs overlapping G4s was calculated as the sQTL enrichment divided by the SNP enrichment. The confidence intervals were calculated in R with geom_smooth using the method "fit = loess".

### Experimental validation of RNA G4 candidates

*NMM ligand enhanced fluorescence*. This assay was performed similarly to our previous work[71]. Briefly, sample solutions containing 1 μM RNA were prepared in 150 mM LiCl/KCl, 10 mM LiCac buffer (pH 7.0) and 1 μM NMM ligand. Fluorescence spectroscopy was performed using HORBIA FluoroMax-4 and a 1-cm path length quartz cuvette (Wuxi Jinghe Optical Instrument Co.) was used with a sample volume of 100 μl. Before the measurement, the samples (ligand not added)

were denatured at 95 °C for 3 min and allowed to cool down at room temperature for 15 min. The samples were excited at 394 nm and the emission spectra were acquired from 550 to 750 nm. Data were collected every 2 nm at 25 °C with 5 nm entrance and exit slit widths. Raw ligand enhanced fluorescence spectra were first blanked with the corresponding sample spectra that resembled all chemical conditions except for the absence of the ligand. All calculations mentioned were performed in Microsoft Excel.

*ThT ligand enhanced fluorescence*. This assay was performed similarly to our previous work[71]. Briefly, sample solutions containing 1 μM RNA were prepared in 150 mM LiCl/KCl, 10 mM LiCac buffer (pH 7.0) and 1 μM ThT ligand. Fluorescence spectroscopy was performed using HORBIA FluoroMax-4 and a 1-cm path length quartz cuvette (Wuxi Jinghe Optical Instrument Co.) was used with a sample volume of 100 μl. Before the measurement, the samples (ligand not added) were denatured at 95 °C for 3 min and allowed to cool down at room temperature for 15 min. The samples were excited at 425 nm and the emission spectra were acquired from 440 to 700 nm. Data were collected every 2 nm at 25 °C with 5 nm entrance and exit slit widths. Raw ligand enhanced fluorescence spectra were first blanked with the corresponding sample spectra that resembled all chemical conditions except for the absence of the ligand. All calculations mentioned were performed in Microsoft Excel.

*Circular dichroism (CD) spectroscopy*. This assay was performed similarly to our previous work[72]. Briefly, the CD spectroscopy was performed using Jasco J-1500 CD spectrophotometer and a 1-cm path length quartz cuvette (Hellma Analytics) was employed in a volume of 2 ml. Samples with 5 μM RNA (final concentration) were prepared in 10 mM LiCac (pH 7.0) and 150 mM KCl/LiCl. Each of the RNA samples was then thoroughly mixed and denatured by heating at 95 °C for 5 min and cooled to room temperature for 15 min for renaturation. The RNA samples were excited and scanned from 220–310 nm at 25 °C and spectra were acquired every 1 nm. All spectra reported were an average of 2 scans with a response time of 0.5 s/nm. They were then normalised to molar residue ellipticity and smoothed over 5 nm. All data were analysed with Spectra Manager™ Suite (Jasco Software).

*Thermal melting monitored by UV spectroscopy*. This assay was performed similarly to our previous work[72]. Briefly, samples were prepared to a concentration of 10 mM LiCac buffer, 150 mM salt (KCl/LiCl) and 5 μM RNA, with a total volume of 2 ml. Each of the samples was mixed thoroughly and heated at 95 °C for 5 min so as to denature the RNA. It was then cooled for 15 min at room temperature for renaturation. All UV-melting experiments were performed on an Agilent Cary 100 UV-Vis Spectrophotometer, using 1-cm path length quartz cuvette. Before the experiment started, the sample block was first flushed with dry N₂ gas and cooled down to 5 °C for 5 min. After the sample solutions were loaded to the cuvettes, they were sealed with 3 layers of Teflon® tape to prevent vaporisation at high temperatures. The samples were scanned from 5 to 95 °C with a temperature incremental rate of 0.5 °C/min. The temperature was held at 95 °C for 5-min before a reversed scan was performed, scanning from 95 to 5 °C with a rate of 0.5 °C/min. The unfolding and folding transitions in both scans were monitored at 295 nm.

Raw data obtained were subtracted by the blank solutions, which contain the same concentrations of LiCac buffer (pH 7.0) and corresponding salt only. It was then smoothed over 11 nm and its first derivative was plotted in Microsoft Excel. The final melting temperature was obtained by averaging the melting temperatures in the forward and reversed scans.

*Intrinsic fluorescence spectroscopy*. This assay was performed similarly to our previous work[72]. Briefly, intrinsic fluorescence spectroscopy was performed using HORIBA FluoroMax-4 and a 1-cm path length quartz cuvette (Hellma Analytics) was used with a volume of 2 ml. Samples with 5 μM RNA were prepared in 10 mM LiCac (pH 7.0) and 150 mM KCl/LiCl. The samples were then denatured at 95 °C for 5 min and cooled to room temperature for 15 min for renaturation. For the measurement of intrinsic fluorescence of G-quadruplexes, the samples are excited at 260 nm and the emission spectra were acquired from 300–500 nm. Spectra were acquired every 2 nm at 25 °C. The bandwidth of the entrance and exit slits was 5 nm. All data were smoothed over 5 nm. Results here are analysed using Microsoft Excel.

**Minigene experiment**. Minigenes were designed using the pI12 splicing reporter sequence[112] and the sequences studied were inserted between the XhoI and XbaI restriction sites. For NAV2, the sequence (hg38 chr11:20,051,051–20,051,629) containing the alternative exon studied and 238 bp upstream of the exon and 296 bp downstream of the exon was selected. For SLC6A17, the sequence (chr1:110,191,667–110,192,950) was selected, which includes 305 bp upstream of exon 7, exon 7 intron 7, exon 8, and 252 bp downstream of exon 8. The minigene sequences were synthesised and cloned into the pTwist-CMV vector from Twist Bioscience using the NotI and NheI sites. For both vectors, we generated a mutant version of RNA G4, NAV2 RNA G4 ggggggtttgggctgggctgggg mutated to gagagtttgagctgagctgagg and SLC6A17 RNA G4 gggcaggggtggggg mutated to gagcagaggtgagag. The full sequences incorporated into NAV2 and

SLC6A17 minigenes are as follows (with nucleotides in upper case letters corresponding to exons):

>NAV2_minigene
TAATACGACTCACTATAGGGAGACCCAAGCTACGTTGGTACCGAGC
TCGGATCCACTAGTAACGGCCGCCAGTGTGCTGGAATTCGAGCTCACT
CTCTTCCGCATCGCTGTCTGCGAGGTACCCTACCAGgtgagtatggatccctctaaa
agcgggcatgacttctagacaggcctcctggtgacctggggtaaagtatgcgctggtgtcagctcaggctg
aggattgggtttctttgtttttccacggatgtggatgtgttgcattgcacgcctagctggataaggcacttcctggtgatgtg
cacctctttctccagggcccttcagtvccctccctagctttccctctctctgccttctgtgtgctgctctgaagttctta
tttttgtttttaacttttcctacagTGACCCGCACCTTGATAGGAACACTTTGCCTAAGA
AAGGACTCAGgtatctgtgtttcctccttgcatctgtgccatctgttgtgtggctttggagcttggctgtgtgactcctt
catggctggtgggggtttgggctgggctgggggtccccgctttgaccaccacagcaggacctttggatgacggctcc
ccttgcaccctctcgttctcactctccatttgtcagcttatttgcttgagcaggggcgtgtgcttttttcaggcttaatgtgg
taaaaccatctcatgaaaaacatccctgggcaagcccaaggagcagtcattactgcttctggggccaatgctcga
gggcgtactaactgggcccttttccctttttttttttcctcagGTCGCGGTTGAGCTGCAGGACAAA
CTCTTCGCGGTCTCTATGCATCCTCCGAACGCCAAGAGCCTAAGCTTAC
TAGAGGGCCCTATTCTATAGTGTCACCTAAAT

>SLC6A17_minigene
TAATACGACTCACTATAGGGAGACCCAAGCTACGTTGGTACCGA
GCTCGGATCCACTAGTAACGGCCGCCAGTGTGCTGGAATTCGAGCTC
ACTCTCTTCCGCATCGCTGTCTGCGAGGTACCCTACCAGgtgagtatggatccctc
taaaagcgggcatgacttctagagttccttagtatgatggcgtcatggggaagccatggtgagcagttaaattactatgtaag
cactcagcatttatgttgtcttaccaagtcctcacagtcatcctctgaagttgatactcttgtgatctgcgtattggggaaact
gaggctcagaggagtagggaaattgcccaagttcacacagcttcttaccactatgggctgctgcctctgaactctgtt
gaggctgagaaaggggggtgtgcacatgaataaaaccatcccctctgtgtctctctttcttcctcctgccttggtcttctgccat
agCTGGACAAGATGCTGGACCCCCAGGTGTGGCGGGAGGCAGCTACCC
AGGTCTTCTTTGCCTTGGGCCTGGGCTTTGGTGGTGTCATTGCCTTCTC
CAGCTACAATAAGCAGGACAACAACTGCCACTTCGATGCCGCCCTGG
TGTCCTTCATCAACTTCTTCACGTCAGTGTTGGCCACCCTCGTGGTGT
TTGCTGTGCTGGGCTTCAAGGCCAACATCATGAATGAGAAGTGTG
TGGTCGAgtaggtggcatctctcctcctgtccctccttctccctgtctaccttacctgggagtgggcaggg
gtggggggcgcaggtgtgcatggggagagaggtcccctccactcagactggaatgagaatcagaggagcact
ctctgtccccagctccgggccacagggacaagctcagagatgcctctgtcagtgacccatgaggttcccacctg
ggtgcctgggaagagcctccaggatctcacccattgcccaccccctgccttcttacctggtcctctcggttttgtgctgcag
GAATGCTGAGAAAATCCTAGGGTACCTTAACACCAACGTCCTGAGC
CGGGACCTCATCCCACCCCACGTCAACTTCTCCCACCTGACCACAAAG
GACTACATGGAGATGTACAATGTCATCATGACCGTGAAGGAGGAC
CAGTTCTCAGCCCTGGGCCTTGACCCCTGCCTTCTGGAGGACGAGCTGG
ACAAGGtgtcgggggacaggctgcccttcccaggacaggcaggaacccagagagcagctgtggccggcggga
gcttgggctcaggcctcaggatgctgacaggtagtcattagtttacttggtaagcaaggatctgctgtgtgtgtccagaggg
agtgaaagggaagaaaggtattggccaaagtccctgcccagaggtaggcttgagcctagacaagaagtagggcagaca
cacacctctcagaagtcacagtaagtgtactcgagggcgtactaactgggcccttttccctttttttttttcctcagGTCGCG
GTTGAGCTGCAGGACAAACTCTTCGCGGTCTCTATGCATCCTCCGAA
CGCCAAGAGCCTAAGCTTACTAGAGGGCCCTATTCTATAGTGTCACCT
AAAT

**Cell transfection.** All plasmids were transfected at 2.5 μg using Fugene HD or lipofectamine 3000 according to the manufacturer's recommendations into DU145 prostate cancer cell line seeded in a 6-well plate.

**Total RNA extraction.** Samples were harvested in TRIsure (Bioline, #BIO-38033). Following the aqueous phase collection, total RNA was isolated using RNA Clean & Concentrator-5 kit (Zymo Research, #R1014) according to the manufacturer's guidelines.

*DNAse treatment of RNA samples.* Total RNA samples were DNAse-treated treated using a TURBO DNA-free™ Kit (Ambion #AM1907) for 30 min at 37 °C and according to the manufacturer's instructions.

*Reverse transcription.* In total, 250 ng of DNAse-treated RNA were reverse transcribed with random hexamers (Invitrogen #N8080127) using SuperScript III™ Reverse Transcriptase (Invitrogen #18080093) in a final volume of 20 μl at 50 °C for 1 h. Control reactions lacking the enzyme were systematically run in parallel as negative controls.

*Semi-quantitative PCR.* Semi-quantitative PCR reactions were performed on 1 μl of diluted RT samples (1/2) with primers PI12_F: GCTCACTCTCTTCCGCATC and PI12_R: CTTGGCGTTCGGAGGATG using One Taq$^R$ DNA polymerase (New England BioLabs #M0486S) and run on a thermocycler with the following conditions: (1) Initial denaturation 2' at 94 °C; (2) {26 cycles} D 30" at 94 °C, A 30" at 58 °C, E 30" at 68 °C; (3) Final extension 4' at 68 °C. PCR products were immediately loaded on a medium-sized 2% agarose gel pre-stained with Ethidium Bromide at a final concentration of 0.5 μg/ml, and run at 80 V for 1 h. Images were acquired on a BioRad Gel Doc system with exposure optimised for "faint bands" and ensuring not to overexpose the signal.

*Cloning and Sanger sequencing of PCR products.* PCR amplicons were excised from the gel, extracted using a ZymoClean™ Gel Recovery Kit (Zymo Research

#D4001), cloned into a TOPO vector (TOPO™ TA Cloning Kit for sequencing #450030) and Sanger-sequenced at Genewiz using a T7 fwd primer.

**Reporting summary.** Further information on research design is available in the Nature Research Reporting Summary linked to this article.

## Data availability

The data of this manuscript have been uploaded to Zenodo with https://doi.org/10.5281/zenodo.6324564.

## Code availability

All the associated code used for the generation of figures and presentation of data throughout the manuscript is deposited on GitHub at the following link: https://github.com/hemberg-lab/Georgakopoulos_Soares_and_Parada_2022.

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

## Acknowledgements

We thank Christopher W.J. Smith for important discussions and critical feedback on the manuscript as well as Omer Ziv for relevant discussion over the design of minigene experiments. I.G.S., G.E.P. and M.H. are supported by the Wellcome Sanger Institute core grant. M.H. is also supported by startup funding from the Evergrande Center. H.Y.W. and C.K.K. are supported by the Shenzhen Basic Research Project [JCYJ20180507181642811]; Research Grants Council of the Hong Kong SAR, China Projects [CityU 11100421, CityU 11101519, CityU 11100218, N_CityU110/17]; Croucher Foundation Project [9509003]; the State Key Laboratory of Marine Pollution Director Discretionary Fund; City University of Hong Kong projects [7005503, 9667222, 9680261]. Additionally, this work was supported by Cancer Research UK (C13474/A18583, C6946/A14492 to E.A.M.) and the Wellcome Trust (104640/Z/14/Z, 092096/Z/10/Z to E.A.M.).

## Author contributions

I.G.S. and G.E.P. conceived the study and carried out the computational analysis, supervised by E.A.M. and M.H. H.Y.W. carried out the G4 validation experiments and data analysis, supervised by C.K.K. The minigene experiment design was conceived by Ro.M. and subsequently performed by Ra.M. and G.F., while supervised by Ro.M., G.E.P. and E.A.M. I.G.S., G.E.P. and M.H. led the writing of the manuscript with input from Ro.M., H.Y.W., C.K.K. and E.A.M.

## Competing interests

I.G.S. and M.H. are founders of Neomer Diagnostics. E.A.M. is a founder and director of STORM Therapeutics. The remaining authors declare no competing interests.
