## [Peer Review File · Nature Communications]

Title: Alternative splicing modulation by G-quadruplexesReviewers' comments:

Reviewer #1 (Remarks to the Author):

In this manuscript, the authors theoretically analyzed the regulatory mechanisms of G4 on alternative splicing. Although the obtained results are interesting, the reviewer believe that the manuscript can be further improved. My concerns are as following.

RNA splicing include five basic modes, namely Exon skipping , Mutually exclusive exons, Alternative donor site, Alternative acceptor site, and Intron retention. The authors are suggested to analyze whether the results in each of these modes.

Reviewer #2 (Remarks to the Author):

Georgakopoulos-Soares et al present with an easy to read manuscript. The introduction is well written and present the field in a refreshing way. The authors identified the G4 structure to be enriched in introns close to the exon-intron junction in both 3' and 5' SS. In addition long G4s were found to have stronger correlation to being at SS. In addition G4 were found to be enriched in short introns and introns that are adjacent to weak SS. Georgakopoulos-Soares analysed AS following stimuli and identified 3 genes with G4 motifs close to AS exon. The authors detected the G4 structure on the RNA level. No validation of the splicing events were performed. Genome wide investigation found a core exon skipping and alternative promoter usage.

Georgakopoulos-Soares describe glorious correlations involving G4 with AS. This connection was described before including potential splicing factors and specific examples. Here the correlations are genome wide and extend to other vertebrates. The manuscript is interesting but the novelty is lacking. In order to make this more novel the authors need to delve into the mechanism in a genome wide manner.

1) Specifically, how does G4 cause core exon skipping?

- What are the splicing factors that are differentially expressed following the stimuli?
- When the splicing factor is KD will you observe the same AS pattern? KD the splicing factor and conduct RNA-seq in the same cells as in the stimuli. What is the overlap between those events and the ones changing following the stimuli with G4 present?
- Do the splicing factor bind the G4 sequence?

2) How many AS events identified had G4 motifs in their flanked introns? How did you chose SLC6A17, UNC13A and NAV2? Where they the top hits? This part is not clear and missing. In addition validation of the AS is needed.

3) is there a connection between the splicing event (inclusion/ exclusion) and the location of the 4G (3' or 5' SS).

Minor comments:

- In general do 4G more abundant in introns? (while correcting for length).
- A short explanation on potassium, Pyridostatin (PDS), Na⁺-K⁺ and Na⁺ 135 -PDS effect on 4G is missing.

Reviewer #3 (Remarks to the Author):

Alternative splicing is crucial in gene expression, which is tightly regulated by many cis- sequence and structural motifs on transcripts. Georgakopoulos-Soares et. al. discovered that a special class of structures, called G-quadruplexes (G4s), are enriched in splicing sites. The study is very interesting in that it points to a new type of RNA structural elements associated with alternative splicing. And more nicely, the study not only performed many correlation analysis of enrichment of G4s in specific splicing sites on genome-wide scale, it did also investigated and validated the association of some G4s with dynamic changes of splicing in response to stimuli. Also, the author found that G4 enrichment at spliced sites is restricted to mammals and birds. However, the main drawback of the study is the lack of proof whether G4s is a driving factor of alternative splicing.

Major points:

- 1, the study Vicki Chambers, 2015, nature biotechnology have already reported a high density of G4 around splice sites. The analysis throughout the study tries to include both DNA-level and RNA-level G4s by considering both template and non-template strands. However, the authors should be cautious in every place they are presenting G4, whether it could be a DNA-G4 or an RNA-G4, or both. The regulation could be very different for a DNA-G4 and an RNA-G4. Except for the dynamic splicing section, it looks like all experimental evidence are just DNA-G4. I recommend to carefully distinguish the use of DNA-G4 and RNA-G4 throughout the manuscript. Considering the study of Chambers 2015, the contribution of this study should be carefully described.
- 2, As the above mentioned, the study are mainly correlation analysis. It would be very convincing to show the deletion/disruption/insertion of a G4, for example those in SLC6A17, UNC13A, and NAV2, would affect alternative splicing. This is not very difficult, if the authors can construct a mini gene that contains the alternatively spliced exons.
- 3, in Figure 1, does the enrichment of G4s are simply due to the GC content variations? I would guess no because of the results from the analysis of the number of G-runs. But it would be nice to just control the GC content to double-check. For example, authors could plot the GC content around the splicing sites and check the enrichment of G4s with respect to different levels of GC content.
- 4, It would be interesting to investigate the association of different protein bindings with the G4s. There are plenty of CLIP datasets available, the authors may be able to identify some associations, which could be informative to define and prove the causality beyond correlation.
- 5, in figure 1B and S1B, why the enrichment in tracks of G4-seq lower than G4 motif? Isn't G4-seq more accurate to predict real G4?
- 6, what is the reason of G4 enrichment in the template strand?

7, do splicing sites of short introns contain more pyrimidine?

8, in figure 3C, why the depletion in upstream introns of length of 2322 and in downstream introns of length of 343? Is this due to small number variation? The authors should include the number of introns in the heatmap.

Minor points:

1, the author should keep the same way in referring to upstream/downstream or 5'/3' splice sites. This becomes a bigger problem in Figure 3E, 5D, I am not sure whether the authors correctly label 5'/3' splice sites.

2, also, the arbitrary using of "enrichment" and "density" in figures 1, 2, 3, 5 (y-axis) is very annoying. It makes the comparison in different benchmark systems impossible. For example, in figure 1A and B, and many other places.

3, the authors should briefly mention the consequence of PDS and K+ treatments on G4 formation to facilitate reading.

4, line 142 "nearby" => how far away?

5, around line 233, the authors should present the results in detail with figures.

6, in line 203: "in accordance with our earlier findings". What're the authors' previous findings?

Reviewer #1

In this manuscript, the authors theoretically analyzed the regulatory mechanisms of G4 on alternative splicing. Although the obtained results are interesting, the reviewer believe that the manuscript can be further improved. My concerns are as following.

RNA splicing include five basic modes, namely Exon skipping , Mutually exclusive exons, Alternative donor site, Alternative acceptor site, and Intron retention. The authors are suggested to analyze whether the results in each of these modes.

We thank the reviewer for the overall positive assessment of our work. We already had the analysis of different types of alternative splicing involved in KCl-induced depolarisation responses, included in Figure 5a and in Supplementary Figure 9 we also provided an analysis of all five basic modes. As requested, we have added a comprehensive analysis of different categories of splicing events from VASTdb (Figure 1d and Supplementary Figure 1e-g). We provide below the paragraph describing the results as well as the new figures and materials and methods.

Figure 1: Non-B DNA motifs at splicing junctions. D. Distribution of G4 motifs relative to cassette exons, alternative acceptor and alternative donor. The orange color in the schematic represents the alternatively included exonic part, corresponding to each type of alternative splicing event.

G4 distribution patterns across splice site categories

We extracted five different types of splicing sites (exon skipping, intron retention, alternative donor, alternative acceptor and mutually coordinated events) from VastDB (Tapial et al. 2017), a curated alternative splicing database. We analyzed the enrichment profile of G4s across the different types of alternative splicing events, and we found distinct differences (Fig 1d, S1e-g). Cassette exons are the most common type of alternative splicing event (Bradley et al. 2012) and we found an enrichment profile for G4s consistent with our previous results (Fig 1a,d). Interestingly, for both alternative acceptor and alternative donor

events, we found that the enrichment was higher for the proximal than for the distal sites (Fig 1d). These results provide evidence that G4s are associated with multiple splice site categories.

Part of supplementary figure 1 is provided below and the relevant figure legend:

Supplementary Figure 1: Distribution of G4s in the vicinity of splice sites. E. Distribution of G4 motifs relative to intron retention sites. F-G. Distribution of G4 motifs relative to mutually coordinated events.

The new methods section for this analysis is provided below:

“We used in-house scripts to process a bed file containing all annotated alternative events which was obtained from VastDB’s UCSC Genome Browser Track Hub (Tapial et al. 2017). We extracted the splice site coordinates from exon skipping (HsaEX), alternative acceptor (HsaALTA), alternative donor (HsaALTD), intron retention (HsaINT) and mutually coordinated events (MULTI), to analyze each category separately for G4 enrichment.”

Reviewer #2

Georgakopoulos-Soares et al present with an easy to read manuscript. The introduction is well written and present the field in a refreshing way. The authors identified the G4 structure to be enriched in introns close to the exon-intron junction in both 3' and 5' SS. In addition long G4s were found to have stronger correlation to being at SS. In addition G4 were found to be enriched in short introns and introns that are adjacent to weak SS. Georgakopoulos-Soares analysed AS following stimuli and identified 3 genes with G4 motifs close to AS exon. The authors detected the G4 structure on the RNA level. No validation of the splicing events were performed. Genome wide investigation found a core exon skipping and alternative promoter usage. Georgakopoulos-Soares describe glorious correlations involving G4 with AS. This connection was described before including potential splicing factors and specific examples. Here the correlations are genome wide and extend to other vertebrates. The manuscript is interesting but the novelty is lacking. In order to make this more novel the authors need to delve into the mechanism in a genome wide manner.

We thank the reviewer for the overall positive assessment of our manuscript. We have now added additional analyses to increase the novelty of our manuscript. We have expanded our analyses to include other types of alternative splicing events by analyzing different categories of splicing events from VASTdb (Figure 1d and Supplementary Figure 1e-g). The relevant figures and text are:

Figure 1: Non-B DNA motifs at splicing junctions. D. Distribution of G4 motifs relative to cassette exons, alternative acceptor and alternative donor. The orange color in the schematic represents the alternatively included exonic part, corresponding to each type of alternative splicing event.

G4 distribution patterns across splice site categories

We extracted five different types of splicing sites (exon skipping, intron retention, alternative donor, alternative acceptor and mutually coordinated events) from VastDB (Tapial et al.

2017), a curated alternative splicing database. We analyzed the enrichment profile of G4s across the different types of alternative splicing events, and we found distinct differences (Fig 1d, S1e-g). Cassette exons are the most common type of alternative splicing event (Bradley et al. 2012) and we found an enrichment profile for G4s consistent with our previous results (Fig 1a,d). Interestingly, for both alternative acceptor and alternative donor events, we found that the enrichment was higher for the proximal than for the distal sites (Fig 1d). These results provide evidence that G4s are associated with multiple splice site categories.

Part of supplementary figure 1 is provided below and the relevant figure legend:

Supplementary Figure 1: Distribution of G4s in the vicinity of splice sites. E. Distribution of G4 motifs relative to intron retention sites. F-G. Distribution of G4 motifs relative to mutually coordinated events.

In addition, our new results provide novel insights about the direct effects of G4s on alternative splicing and how RBPs mediate the impact. We have also analyzed sQTLs from the GTEx consortium, as well as eCLIP experiments for all available RBPs from the ENCODE consortium. Moreover, we have analyzed loss of function experiments, in which RBPs were knocked down or knocked out. We found enrichment of sQTLs at G4 sites flanking splice sites and we also found specific RBPs that interact with G4 sites to modulate splicing. These analyses provide further evidence that G4s can directly modulate alternative splicing and expand our understanding of the underlying molecular mechanisms. Finally, we have also performed a minigene experiment, verifying directly that mutations of sequences that lead to G4 formation in RNA (RNA G4) affect splicing. The combination of the large-scale analyses of high-throughput experiments from GTEx and ENCODE along with the minigene experiment greatly improves the novelty of our manuscript. Crucially, they allow us to draw stronger conclusions as we are no longer limited to drawing inferences based on correlative observations.

1) Specifically, how does G4 cause core exon skipping?

- What are the splicing factors that are differentially expressed following the stimuli?

We have carried out differential expression analysis of splicing factors in response to KCl treatment (Response Figure 1). The significant hits include genes coding for two spliceosomal proteins (GEMIN5 and SMNDC1) and 7 involved in splicing regulation (TIA1, PCBP2, SRSF1, AKAP8L, TIAL1, TARDBP and RBM15). However, the magnitude of these gene expression changes is arguably small and given the relatively short interval between the time points for the RNAseq samples, we find it most likely that the changes in splicing are due to post-translational effects rather than altered expression of splicing regulators. To clarify this point we included the following statement in the discussion:

We observed widespread exon skipping following potassium depolarisation of neurons (Fig 5a, S8), a phenomenon that to our knowledge has only been documented for a handful of cases (An and Grabowski 2007), (J.-A. Lee et al. 2007), (Liu et al. 2012), (Fiszbein and Kornblihtt 2017). These alternative splicing changes are likely induced by Ca^{2+} influx after depolarisation that is known to affect splicing via CaMK IV (Sharma and Lou 2011). Here, we show that the changes in splicing patterns are associated with the presence of G4s at the splice junctions. **Given the relatively short interval between the time points for the RNAseq samples, we find it most likely that the changes in splicing are due to post-translational effects rather than altered expression of splicing regulators. In fact, part of the alternative splicing changes observed in response to depolarization has been shown to be dependent on hnRNP L phosphorylation by CaMK IV (Liu et al. 2012).**

Response Figure 1: Differentially expressed splicing factors after KCl-induced depolarisation. Volcano plots showing differential gene expression results after KCl-induced depolarisation for a selection of genes annotated as spliceosomal proteins (A) or splicing regulatory factors (B). Orange dots represent significant changes with adjusted p-values < 0.05 . The red dots represent absolute $\log_2(\text{fold-change}) > 0.5$ and significant changes with adjusted p-values < 0.05 . All the genes included in this selection were part of the factors for which we analysed eCLIP and knock-down RNA-seq data (LoF experiments).

- When the splicing factor is KD will you observe the same AS pattern? KD the splicing factor and conduct RNA-seq in the same cells as in the stimuli. What is the overlap between those events and the ones changing following the stimuli with G4 present?

We thank the reviewer for their comment. We analyzed the differential binding of RBPs at splice sites that are flanked by G4 sequences using eCLIP data for K562 and HEPG2 cell lines from the ENCODE consortium (Van Nostrand et al. 2020). In total, we examined 1,346 experiments of 150 RBPs. 722 experiments were performed in K562 and 624 experiments were performed in HepG2, for 120 and 103 RBPs respectively. We found specific clusters of RBPs that are associated with the presence of G4 motifs. The association between splicing factors and G4s was linked to their orientation and their positioning in the splice junction, with differences observed for template and non-template orientations and for upstream and downstream splice sites. Moreover, we analysed 506 loss of function (LoF) experiments followed by RNA-seq that targeted a total of 269 RBPs in HepG2 and K562 cell lines. Differential splicing analyses of these RNA-seq experiments enabled us to identify factors that were significantly associated with G4 flanked exons. The new figure 6 now encompasses the full RBP analysis. We have also added the following paragraphs:

Systematic characterization of the interplay between RBPs and G4s at splice sites

During gene expression, RNA-binding proteins (RBPs) enable different catalytic steps of RNA processing and serve as key regulatory factors of alternative splicing. We processed data from 1,345 eCLIP experiments that were performed on K562 and HepG2 cell lines to calculate differential binding of RBPs between exons that are flanked and exons that are not flanked by G4s within a 200 bp window. We performed unsupervised hierarchical clustering of RBPs based on their G4 enrichment profile, taking into consideration G4 motifs at the template and non-template orientations in proximity to splice sites. Using unsupervised hierarchical clustering we identified a total of ten clusters with distinct differential binding patterns (Fig 6a), of which all except cluster 6 showed substantial and significant enrichment differences between exons that are flanked by G4 motifs and exons that are not flanked by G4s. However, the other constituent clusters exhibited clearly distinguishable patterns; for instance, in cluster 7 we observed an enrichment difference only for G4s found at the non-template orientation, whereas for cluster 10 we observed an enrichment difference specific to the template orientation. These results show that several RBPs differentially bind to splice sites flanked by G4s.

To complement our observations from eCLIP data, we analysed 506 loss of function (LoF) experiments followed by RNA-seq that targeted a total of 269 RBPs in HepG2 and K562 cell lines, from which 143 RBP factors overlapped between the eCLIP and LoF experiments. We performed quantitative analyses to determine the number of differentially included exons induced by the LoF of target RBPs. We found that for 36 RBPs, differential inclusion following the loss of function of the RBP is associated with the presence of a G4 motif in proximity to the splice site in at least one of the analyzed LoF experiments (chi-squared test, adjusted p-value < 0.05) (Fig 6b). Integrating eCLIP with LoF RNA-seq experiments we obtained a high confidence list of 15 differentially bound RBPs to G4 sites that show a significant and consistent association with alternative splicing (Fig 6b). Interestingly, we

found examples such as HNRNPK and HNRNPU, which exhibit higher binding to G4 around splice sites and are positively associated with differentially included exons (p -value <0.05 and $\log_2(\text{OR}) > 0$), suggesting that these factors could have a direct impact on G4-mediated AS regulation (Fig 6c). Conversely, we also found examples such as AQR and RBM15, which are depleted around splice sites flanked by G4 and are negatively associated differentially included exons (p -value <0.05 and $\log_2(\text{OR}) < 0$), suggesting that binding and impact of these factors over AS could be prevented by G4 forming motifs (Fig 6c). We also found cases such as RBFOX2, that exhibit remarkably different binding profiles across splice sites flanked by G4, although they were not positively or negatively associated with differentially included exons.

Figure 6: The interplay between RNA binding proteins and G4s in proximity to splice sites. **A.** Enrichment difference in the distribution of RBPs in sites with and without G4s is shown for the template and non-template G4 orientations. Heatmap displaying the clustering of differential enrichment values of each RBP between splice sites flanked by non-template (left) or template (right) G4s. **B.** Functional assessment of the RBP effect on alternative inclusion of exons flanked by G4s. Each dot represents a single replicate performed for an RBP on a given cell line. Log(odds ratio) of differential inclusion for sites with and without G4s, following RBP KO experiments. Significant associations between RBPs and alternative inclusion of G4-flanked exons are highlighted either in green (HepG2) or purple (K562). Labeled factors are the set of high-confidence factors identified by consistent eCLIP and

LoF followed by RNA-seq analyses. Results are shown in aggregate, as well as for template and non-template orientations separately. Statistical evaluation was performed using chi-squared tests with Bonferroni corrections. Results for both eCLIP and RNA-seq experiments that target the same factor, are shown side by side. **C.** Enrichment of a panel of RBP binding sites across splice sites flanked or not flanked by G4s. Displayed factors are AQR, RBM15, HNRPK, HNRPU and RBFOX2, each of which belongs to a different eCLIP cluster. The error bands represent 95% confidence intervals based on a binomial model.

We have also added the new methodology in the materials and methods section which we also provide below:

4.4. Integration of eCLIP and LoF followed by RNA-seq experiments.

eCLIP data analysis. eCLIP data for K562 and HepG2 cell lines were derived from the ENCODE consortium (Van Nostrand et al. 2020), including 1,346 experiments of 150 RBPs. Among them, 722 experiments were performed in the K562 cell line and the remaining 624 experiments were performed in the HepG2 cell line, for 120 and 103 RBPs respectively. To investigate the relationship between G4 sites, we extracted splice sites flanked by G4 within 100-bp intronic windows, with either template or non-template orientation. Splice site regions were then separated into 10bp bins. For each bin we calculated the factor binding enrichment across the different groups of splice sites (flanked by template or non-template G4s, or not flanked by G4s), and we then calculated the differential enrichment values between splice sites flanked and not flanked by G4s. We assessed the statistical significance of these differences using chi-squared testing with Bonferroni correction. For clustering based on differential enrichment, non-significant differences were set to 0. Differential enrichment values of template and non-template G4 bins were clustered using Ward's method with unsupervised hierarchical clustering with the hclust package in R to classify RBP enrichment profiles into 10 clusters.

Analysis of LoF experiments followed by RNA-seq. We analysed 506 LoF experiments followed by RNA-seq that targeted a total of 269 RBPs in HepG2 and K562 cell lines, all of which were derived from the ENCODE consortium (Van Nostrand et al. 2020). To quantify alternative splicing changes after knockout or knockdown of target RBPs, we used Whippet (Sterne-Weiler et al. 2018), which led us to obtain sets of differentially included exons associated with each LoF condition. Similarly to the KCl-induced depolarization experiment analyses, we used the absolute value of $\Delta\text{PSI} > 0.1$ and probability > 0.9 to define a splicing node as differentially included between treatments and controls. For each LoF experiment, we calculated the association between G4s and differentially included exons estimated as log odds ratio and assessed the statistical significance using chi-squared test with Bonferroni correction. To get a list of high-confidence factors that have a role in G4-mediated alternative splicing, we considered factors that were found to have significant eCLIP and RNA-seq analyses performed for the same cell line (K562 or HepG2). Finally, to check for experimental consistency, only factors with at least two eCLIP replicates that clustered in the same group were considered to be high confidence and they were labelled in Figure 6b.

- Do the splicing factor bind the G4 sequence?

We have analyzed eCLIP-seq data, which indicated that specific splicing factors bind at G4 sequences (see previous point). These are summarized in Figure 6 and include different patterns of binding, e.g. HNRNPU and RBFOX2 have higher enrichment at G4s, while other RBPs such as AQR bind preferentially to sequences without G4 motifs.

- 2) How many AS events identified had G4 motifs in their flanked introns?

We investigated the proportion of AS events with G4 motifs within 100bp within the proximal upstream and downstream introns for the one experiment in human cells and the four experiments in mouse cells. In the human experiment, we found 172 out of 2,362 AS events harbored G4s, whereas 208 out of 2,362 AS events harbored G4s in the downstream 100bp intronic region.

How did you choose SLC6A17, UNC13A and NAV2? Where they the top hits? This part is not clear and missing. In addition validation of the AS is needed.

We apologize for the lack of clarity regarding the selection. To select the candidates for the experimental validation we analysed the full set of available RNA-seq experiments included in the study (Qiu et al. 2016). These corresponded to 2 experiments in human cells and 12 in mouse cells. We used the absolute value of DeltaPSI >0.1 and probability >0.9 to define a splicing node as differentially included between treatments and controls. We found a total of 44 differentially included exons between human and mouse cells in at least four different experiments. From these, we selected SLC6A17, UNC13A and NAV2, because they have non-template G4 motifs that have the potential to form in the RNA. We have updated the methods section and the results on p9 to better explain the selection.

- 3) is there a connection between the splicing event (inclusion/ exclusion) and the location of the 4G (3' or 5' SS).

We thank the reviewer for their comment. We tested the enrichment of upstream and downstream G4s with exon exclusion, and in both cases, we observed a significant association (p-values < 1×10^{-15}). However, the odds ratios representing upstream and downstream associations to exon exclusion events did not have statistically significant differences (odds ratios of 7.24 and 7.41 respectively, p-value>0.05). The same type of analysis was performed for inclusion, but due to the low number of differentially included exons flanked by G4s, we could not reach meaningful conclusions.

Minor comments:

- In general do 4G more abundant in introns? (while correcting for length).

We find a small overall enrichment for G4s at introns after correcting for length. There are 191,890 G4 motifs within introns representing a 1.06-fold enrichment over the

genome average, correcting for length and subtracting non-mappable regions (Chi-square test, p-value<0.05).

- A short explanation on potassium, Pyridostatin (PDS), Na⁺-K⁺ and Na⁺ 135 -PDS effect on 4G is missing.

We thank the reviewer for the comment. We have now updated the text to provide a brief explanation of their effect on G4s.

“PDS, is a highly potent small molecule that binds and stabilizes G4s. PDS, K⁺ and Na⁺ molecules selectively interact with G4s and stabilize them (Bochman et al. 2012). Compared to Na⁺, K⁺ stabilizes G4 assemblies to a larger extent.”

Reviewer #3

Alternative splicing is crucial in gene expression, which is tightly regulated by many cis-sequence and structural motifs on transcripts. Georgakopoulos-Soares et. al. discovered that a special class of structures, called G-quadruplexes (G4s), are enriched in splicing sites. The study is very interesting in that it points to a new type of RNA structural elements associated with alternative splicing. And more nicely, the study not only performed many correlation analysis of enrichment of G4s in specific splicing sites on genome-wide scale, it did also investigated and validated the association of some G4s with dynamic changes of splicing in response to stimuli. Also, the author found that G4 enrichment at spliced sites is restricted to mammals and birds. However, the main drawback of the study is the lack of proof whether G4s is a driving factor of alternative splicing.

We thank the reviewer for the overall positive assessment. We have now performed additional experiments and analyses to further back our claims. These include the analysis of eCLIP experiments for the investigation of RBP binding to G4 sites, loss of function experiments followed by RNA-seq to investigate the association between RBPs that bind to G4 sites and the modulation of splicing events, sQTL enrichment analysis at G4 motifs and an experiment with a minigene in which we studied how mutations of G4 sites influence alternative splicing events. The addition of these new results provide further evidence that G4s directly modulate alternative splicing outcomes.

Major points:

1, the study Vicki Chambers, 2015, nature biotechnology have already reported a high density of G4 around splice sites. The analysis throughout the study tries to include both DNA-level and RNA-level G4s by considering both template and non-template strands. However, the authors should be cautious in every place they are presenting G4, whether it could be a DNA-G4 or an RNA-G4, or both. The regulation could be very different for a DNA-G4 and an RNA-G4. Except for the dynamic splicing section, it looks like all experimental evidence are just DNA-G4. I recommend to carefully distinguish the use of DNA-G4 and RNA-G4 throughout the manuscript. Considering the study of Chambers 2015, the contribution of this study should be carefully described.

We thank the reviewer for the comment and we agree that the distinction between DNA-G4s and RNA-G4s is crucial. We have modified our terminology throughout the manuscript to make this clear. We have also revised the text to further describe the contributions of Chambers et al. 2015:

“G4-seq utilizes the fact that stable G4s can stall the DNA polymerase *in vitro*, thereby allowing high throughput sequencing to be used to detect DNA G4s at high resolution. Chambers et al provided the first method that enabled genome-wide detection of sites with G4 formation potential, and they identified non canonical structural features of G4 formation

as well as regions in the genome that are more likely to harbor G4s, such as such as 5' untranslated regions and splicing sites.”

2, As the above mentioned, the study are mainly correlation analysis. It would be very convincing to show the deletion/disruption/insertion of a G4, for example those in *SLC6A17*, *UNC13A*, and *NAV2*, would affect alternative splicing. This is not very difficult, if the authors can construct a mini gene that contains the alternatively spliced exons.

We thank the reviewer for their comment, we have carried out the requested experiment. We found direct evidence of *SLC6A17* and *NAV2* RNA G4s regulation of splicing. We did not test *UNC13A* since the exon of interest is flanked by multiple G4s that could compensate for the disruption of an individual G4.

“To provide direct evidence for the role of G4s in the modulation of alternative splicing events, we designed two minigene constructs that contain wild-type and mutant G4 motifs, which we previously validated to lead to RNA-G4 structure formation in the *SLC6A17* and *NAV2* genes (Fig 4). Within these constructs we included the whole sequence of exons that were differentially included after KCl-induced depolarisation and corresponding flanking regions (Fig S12). In the case of *SLC6A17* minigene, we inserted a wild-type or mutated G4 motif flanked by two exons, one of them corresponding to a KCl-responding alternative exon (Fig 5f). In the *SLC6A17* minigene containing the wild-type G4 motif, we observed two main splicing products corresponding to isoforms where either both exons are included or excluded and a minority product where only one alternative exon is included (Fig 5g). After the introduction of mutations in the G4 motif, we observed strong exclusion of both exons. Similarly, for the *NAV2* minigene experiments, we also observed that mutations over the flanking G4 favoured exon skipping (Fig S12). These results indicate that G4s present in flanking intronic regions can have a direct effect over alternative splicing outcomes, favouring exon inclusion, which is in agreement with previous observations (Huang et al. 2017).

To investigate if G4 motifs have a transcriptome-wide effect over alternative splicing, we took advantage of splicing quantitative trait loci (sQTL) data from GTEx consortium data (Consortium and The GTEx Consortium 2020). Since G4s are reported to be enriched in both germline and somatic mutations (Georgakopoulos-Soares et al. 2018; Guiblet et al. 2021), we adjusted the enrichment of sQTLs for differences in SNP distribution across the splice junction. We found that sQTLs were more likely to overlap G4s that are in close proximity to splice sites (Fig 5h). The highest sQTL adjusted enrichment values were found in exonic regions and the most proximate flanking intronic regions. Therefore, we conclude that G4s are enriched for sQTLs relative to the expected population variant frequency, suggesting a functional role..”

We also provide the updated figure and the associated part of the figure legend here which includes the minigene experiment:

“**F.** Schematic showing the design of a minigene assay to test the effect of G4 motifs on alternative splicing. Two consecutive exons from *SLC6A17* and their flanking intronic regions were inserted into the minigene construct. The red exon corresponds to the alternative exon highlighted in C, while the orange exon represents the following downstream exon that was not detected as alternatively included after KCl treatment. Arrow above the G4 motif indicates its location on the non-template strand. Wild-type and mutant G4 sequences are shown with G-runs underlined and mutations in red. **G.** Minigene assay results show the effect of mutations in G4 motifs, which strongly promote the skipping of both *SLC6A17* exons. **H.** Adjusted enrichment of sQTLs at G4s relative to splice sites. The error bands represent 95% confidence intervals based on a binomial model.”

We also provide the supplementary figure below.

Supplementary Figure 12

Supplementary Figure 12: Minigene experiments with wild-type and mutant G4 sequences flanking the splice site of the target exon. A. The schematic on top shows the design, where a NAV2 exon that is flanked by two G4s was inserted into a minigene. The arrows indicate the strand where G4s are present. The G-runs of the downstream nontemplate G4 were mutated to prevent G4 formation. PCR followed by electrophoresis shows that the NAV2 exon (red) is predominantly excluded after mutagenesis. Interestingly, the G4 mutations also activate a cryptic splice site that is not observed in wild-type sequences (green), and it was confirmed by Sanger sequencing. Double bands match the expected weight of NAV2 including isoform. **B-C.** Most expected alternative splicing products were confirmed by Sanger sequencing. Analysis of the sequencing results also corroborate the activation of cryptic splice sites upon the introduced mutations at target G4 sequences. Black and red indicate wild-type and cryptic splice sites respectively. **D-E.** Consistent band migration patterns were observed across replicate experiments that were performed after independent cell transfections (Replicate #1 and #2). PCR replicates also show consistent band migration patterns (top row and bottom row), showing a reproducible effect of mutations over downstream non-template G4s over alternative splicing.

3, in Figure 1, does the enrichment of G4s are simply due to the GC content variations? I would guess no because of the results from the analysis of the number of G-runs. But it would be nice to just control the GC content to double-check. For example, authors could plot the GC content around the splicing sites and check the enrichment of G4s with respect to different levels of GC content.

It has been shown that the G-quadruplex enrichment does not follow the same pattern as GC-content variation relative to the splice sites (Li et al. 2018). In addition, we have performed simulations controlling for mono/di nucleotide content around splice sites, which does not explain the enrichment patterns.

We provide figure 2 from Li et al. 2018 (PMID: 30104384) below:

We have revised the Results section on p5 and we now mention this work.

4, It would be interesting to investigate the association of different protein bindings with the G4s. There are plenty of CLIP datasets available, the authors may be able to identify some associations, which could be informative to define and prove the causality beyond correlation.

We thank the reviewer for their suggestion. We have added a new section and a new figure 6 where we characterize the association between RNA G4s and RBPs. The same point was raised by both Reviewer 1 and 2, and we refer the reviewer to those responses. Briefly, we analyzed all ENCODE CLIP experiments and we found a significant association between G4 sites and multiple RBPs. Moreover, we additionally analysed RNA-seq experiments in two human cell lines after the knockdown or knockout of a collection of RBPs. Some of the

factors that we found associated with G4-mediated alternative splicing were already known to interact with RNA G4s while other factors have not been previously associated with G4s.

5, in figure 1B and S1B, why the enrichment in tracks of G4-seq lower than G4 motif?
Isn't G4-seq more accurate to predict real G4?

We thank the reviewer for their point. Although G4-seq can accurately identify formation of G4 *in-vitro*, the assay has relatively low resolution as the identified peaks are >100bp long. In addition, the conditions of G4-seq (K⁺ /PDS treatment result in substantial differences in the number of G4s identified which contributes to the difference in enrichment observed.

We also provide the peak length for the G4-seq experiments below:

The median peak length for G4-seq peaks from (Chambers et al. 2015) was 255 nt and 195 nt in Na⁺-K⁺ and Na⁺-PDS conditions, respectively. Median peak length for G4-seq peaks from (Marsico et al. 2019) was 135 nt and 120 nt in PDS and K⁺ conditions, respectively.

6, what is the reason of G4 enrichment in the template strand?

An enrichment of G4s at the template strand could implicate G4 formation at the DNA level during co-transcriptional splicing. It could also involve other structures such as the i-motif. Our analyses suggest enrichment in both orientations, albeit at higher levels at the non-template strand. We have now updated the discussion to include this:

“We find an enrichment of G4s at the template strand which suggests formation and potential roles at the DNA level as well. This could occur during co-transcriptional splicing when the DNA is single-stranded or through formation of i-motifs (Zeraati et al. 2018).”

7, do splicing sites of short introns contain more pyrimidine?

We separated introns into two groups, those larger than 500bp and those smaller than 500bp. For each group, we estimated the proportion of polypyrimidines (two or more consecutive) in the non-template strand in 25bp within the intron and found 2.106% versus 2.133% content representing a small but statistically significant difference in polypyrimidine content (Mann-Whitney U test, p-value<0.01).

8, in figure 3C, why the depletion in upstream introns of length of 2322 and in downstream introns of length of 343? Is this due to small number variation? The authors should include the number of introns in the heatmap.

We thank the reviewer for their valuable comment. Indeed, the pattern was not entirely consistent due to small number variation, however we found a mistake from our end that led to only a subset of the annotated events being considered in the analysis. Now we consider all the events and also take into account the reviewer's point to show the number of events in the graph (represented as the size of the dots). With these changes we have improved considerably this figure panel which we provide here and have updated the manuscript

figure accordingly. The conclusions remain the same but now the effects of intron size and splice strength are even more visible.

Minor points:

1, the author should keep the same way in referring to upstream/downstream or 5'/3' splice sites. This becomes a bigger problem in Figure 3E, 5D, I am not sure whether the authors correctly label 5'/3' splice sites.

We apologize for this mistake. We have now corrected Figure 3E and 5D. We have also changed the terminology and use upstream/downstream consistently throughout the figures.

2, also, the arbitrary using of "enrichment" and "density" in figures 1, 2, 3, 5 (y-axis) is very annoying. It makes the comparison in different benchmark systems impossible. For example, in figure 1A and B, and many other places.

We apologize for that. We have now changed this to use enrichment in figure 1A and 1B to make the comparison easier. However, there is the exception of figure 7A,C for which we believe the density also reflects the overall differences between species.

3, the authors should briefly mention the consequence of PDS and K⁺ treatments on G4 formation to facilitate reading.

We thank the reviewer for the comment. We have now changed the text to explain the effect of PDS and K⁺ treatments on G4 formation. Specifically we added the following sentence:

“PDS, is a highly potent small molecule that binds and stabilizes G4s. PDS, K⁺ and Na⁺ molecules selectively interact with G4s and stabilize them (Bochman, Paeschke, and Zakian 2012). Compared to Na⁺, K⁺ stabilizes G4 assemblies to a larger extent. ”

4, line 142 “nearby” => how far away?

We thank the reviewer. We have now explained this better, we referred to 100bp from the splice junction.

5, around line 233, the authors should present the results in detail with figures.

We thank the reviewer for their comment. We have now added the figure S5c to illustrate this which we provide below:

Supplementary Figure 5: **A.** Intron size of the upstream and downstream introns was calculated for groups with or without a G4 within 100 bps of the splice site (Mann-Whitney U p-value <0.001 for both upstream and downstream introns). **B.** GC content distribution across selected groups of short and long introns. Intron size interval refers to the size of small introns. Long introns were defined as introns > 500 bp. **C.** Intron size for introns with template and non-template G4s within 100bp in the intronic region, for upstream and downstream splice sites. **D.** G4 motif enrichment relative to the splice site across exons in the gene body at the 3'ss (upstream) and at the 5'ss (downstream). G4 motifs are enriched at both upstream (3'ss) and downstream (5'ss) across splice sites throughout the gene body. Exons were separated into first to fourth exons, middle exons, last four exons and the

distribution of G4s was studied individually at each of them.

6, in line 203: “in accordance with our earlier findings”. What’re the authors’ previous findings?

We thank the reviewer, we have now removed the statement “in accordance with our earlier findings”.

REVIEWERS' COMMENTS

Reviewer #1 (Remarks to the Author):

The authors have replied all my comments.

Reviewer #2 (Remarks to the Author):

The authors addressed all my concerns. This a very interesting paper and should be accepted for publication.

Reviewer #3 (Remarks to the Author):

I am very happy to see the revision has addressed almost of my concerns. Just three comments:

1, the mini-gene validation is cool. However, the authors may consider a design of restore mutations to see whether it rescues the phenotype. Otherwise, the change of splicing may be the consequence of the changes of sequences.

2, from the manuscript, I understand that every occurrence of "G4s" may be DNA G4s, or RNA G4s, or possibly DNA/RNA G4s (as a genomic sequence pattern). But sometimes it may be misleading if the identification is not clear. I had suggested to explicitly identify these for every occurrence, and I am happy to see much clearer presentation in the revision. However, it is still not enough, I suggest to further identify them at least for every section/paragraph, so that a reader knows what it means. For example, the section title "Sequence analysis and experimental data show that G4s are enriched near splice sites" needs to be DNA G4s.

3, the third paragraph of introduction says "In addition to RBPs, secondary and tertiary structures at both the DNA and RNA level are known to modulate alternative splicing". Although DNA G4 does regulates transcription, it is difficult to understand how DNA G4 regulates splicing. And the reference in this paragraph does not support that.

4, where is figure 5G 5H?

Our responses to the reviewers' points are:

Reviewer #1 (Remarks to the Author):

The authors have replied all my comments.

We thank the reviewer who encouraged us to analyze other types of alternative splicing, which helped strengthen the manuscript.

Reviewer #2 (Remarks to the Author):

The authors addressed all my concerns. This a very interesting paper and should be accepted for publication.

We thank the reviewer for all their constructive inputs and positive feedback.

Reviewer #3 (Remarks to the Author):

I am very happy to see the revision has addressed almost of my concerns.

We thank the reviewer for their thorough revision and we are pleased to know they are positive about our latest work to address their initial concerns.

Just three comments:

1, the mini-gene validation is cool. However, the authors may consider a design of restore mutations to see whether it rescues the phenotype. Otherwise, the change of splicing may be the consequence of the changes of sequences.

We are happy to see that our efforts to incorporate the minigene experiments were valued positively by the reviewer. However, we think that the additional experiments they are suggesting would not disentangle the effect of sequence composition over structure formation. For the minigene assay we synthesized minigene constructs with wild-type and mutated sequences. Consequently, "restoring the mutations" would lead to the same wild-type constructs that we already assayed. Due to these and other limitations of the minigene assays, we complemented these results with sQTL analyses that provide more generalisable evidence to show a functional relationship between G4 and alternative splicing.

2, from the manuscript, I understand that every occurrence of "G4s" may be DNA G4s, or RNA G4s, or possibly DNA/RNA G4s (as a genomic sequence pattern). But sometimes it may be misleading if the identification is not clear. I had suggested to explicitly identify these for every occurrence, and I am happy to see much clearer presentation in the revision. However, it is still not enough, I suggest to further identify them at least for every section/paragraph, so that a reader knows what it means. For example, the section title "Sequence analysis and experimental data show that G4s are enriched near splice sites" needs to be DNA G4s.

We thank the reviewer for their comment. We have revised the sentence to:

"Sequence analysis and experimental data show that DNA G4s are enriched near splice sites"

We have also used this opportunity to further revise and explicitly identify DNA G4s, RNA

G4s and DNA/RNA G4s. We have changed the following paragraph titles:
The paragraph title: "G4 distribution patterns are found across splice site categories"

To

"DNA G4 distribution patterns are found across splice site categories"

The paragraph title: "G4s are preferentially found on the non-template strand"

To

"DNA G4s are preferentially found on the non-template strand"

The paragraph title: " G4s are enriched at weak splice sites"

To

"DNA G4s are enriched at weak splice sites"

The paragraph title: " G4s are enriched for short introns"

To

"DNA G4s are enriched for short introns"

The paragraph title: "Enrichment of G4s at splice sites does not extend beyond vertebrates"

To

"Enrichment of DNA G4s at splice sites does not extend beyond vertebrates"

We have also revised additional G4 statements in the text, found in red.

3, the third paragraph of introduction says "In addition to RBPs, secondary and tertiary structures at both the DNA and RNA level are known to modulate alternative splicing". Although DNA G4 does regulate transcription, it is difficult to understand how DNA G4 regulates splicing. And the reference in this paragraph does not support that.

We thank the reviewer for their comment and we agree that the reference does not support our claim. The new sentence reads:

"In addition to RBPs, secondary RNA structures are known to modulate alternative splicing (Shepard and Hertel 2008), yet little is known about the impact of DNA secondary structures over alternative splicing."

4, where is figure 5G 5H?

We thank the reviewer for pointing this mistake out, the figure panels we were referring to were figure 5D-E. We have now corrected this in the text.